# Capacity to provide care for common childhood infections at low-level private health facilities in Western, Uganda

Juliet Mwanga-Amumpaire[1,2]*, Grace Ndeezi[3], Karin Källander[4,5], Celestino Obua[1], Richard Migisha[1], Juvenal Nkeramahame[6], Cecilia Stålsby Lundborg[4], Joan Nakayaga Kalyango[2,7], Tobias Alfvén[4,8]

1 Mbarara University of Science and Technology, Mbarara, Uganda, 2 Clinical Epidemiology Unit, College of Health Sciences, Makerere University, Kampala, Uganda, 3 Department of Pediatrics and Child Health, College of Health Sciences, Makerere University, Kampala, Uganda, 4 Department of Global Public Health, Karolinska Institutet, Solna, Sweden, 5 Programme Division, Health Section, UNICEF, New York, New York, United States of America, 6 Faculty of Medicine and Health Sciences, University of Antwerp, Antwerp, Belgium, 7 Department of Pharmacy, College of Health Sciences, Makerere University, Kampala, Uganda, 8 Sachs' Children and Youth Hospital, Stockholm, Sweden

* jmwanga@must.ac.ug

## Abstract

### Background

Low-level private health facilities (LLPHFs) handle a considerable magnitude of sick children in low-resource countries. We assessed capacity of LLPHFs to manage malaria, pneumonia, diarrhea, and, possible severe bacterial infections (PSBIs) in under-five-year-olds.

### Methods

We conducted a cross-sectional survey in 110 LLPHFs and 129 health workers in Mbarara District, Uganda between May and December 2019. Structured questionnaires and observation forms were used to collect data on availability of treatment guidelines, vital medicines, diagnostics, and equipment; health worker qualifications; and knowledge of management of common childhood infections.

### Results

Amoxicillin was available in 97%, parental ampicillin and gentamicin in 77%, zinc tablets and oral rehydration salts in >90% while artemether-lumefantrine was available in 96% of LLPHF. About 66% of facilities stocked loperamide, a drug contraindicated in the management of diarrhoea in children. Malaria rapid diagnostic tests and microscopes were available in 86% of the facilities, timers/clocks in 57% but only 19% of the facilities had weighing scales and 6% stocked oxygen. Only 4% of the LLPHF had integrated management of childhood illness (IMCI) booklets and algorithm charts for management of common childhood illnesses. Of the 129 health workers, 52% were certificate nurses/midwives and (26% diploma nurses/clinical officers; 57% scored averagely for knowledge on management of common childhood illnesses. More than a quarter (38%) of nursing assistants had low

**Data Availability Statement:** All relevant data are within the manuscript and its Supporting information files.

**Funding:** The research was funded by Makerere-SIDA (MAK-SIDA) project SE-0-SE-6-5118006001-UGA-43082 SIDA Phase IV Project 346 bilateral research program.

**Competing interests:** The authors have declared that no competing interests exist.

**Abbreviations:** AHPC, Allied Health Professionals' Council; IMCI, Integrated management of childhood illnesses; LLPHF, Low-level private health facility; LMICs, Low and Middle-Income Countries; MoH, Ministry of Health; MUAC, Mid upper arm circumference; NGO, Non-governmental organization; ORS, Oral rehydration salts; PFP, Private for profit; PNFP, Private not for profit; PSBI, Possible severe bacterial infection; RDT, Rapid diagnostic test; UMDPC, Uganda Medical and Dental Practitioners' Council; UNMC, Uganda Nurses and Midwives Council.

knowledge scores. No notable significant differences existed between rural and urban LLPHFs in most parameters assessed.

## Conclusion

Vital first-line medicines for treatment of common childhood illnesses were available in most of the LLPHFs but majority lacked clinical guidelines and very few had oxygen. Majority of health workers had low to average knowledge on management of the common childhood illnesses. There is need for innovative knowledge raising interventions in LLPHFs including refresher trainings, peer support supervision and provision of job aides.

## Introduction

The Universal Health Coverage (UHC) is one of the targets of the Sustainable Development Goal 3(SGD3). It aims at improving health outcomes through expanding access to essential healthcare services for all people in need while minimizing risk of encountering excessive financial hardships [1, 2]. Many African countries are still lagging behind, in achieving the UHC target yet the year 2030 is approaching [3]. In 2019, Sub-Saharan Africa accounted for half of the over 5 million deaths which occurred globally in children below 5 years of age [4]. Uganda has made some progress towards reducing the under-five mortality which now stands at 46 deaths per 1000 live births. This, however, is still above the 25 deaths per 1000 lives target of SDG 3.2 [5]. Most of the deaths result from preventable common childhood illnesses that include malaria, pneumonia, diarrheal illnesses and neonatal conditions which can be managed with good outcomes if quality health care services are accessible to all persons [4–10]. Further, child mortality has been found to be higher in rural communities living far away from health facilities [11].

In the year 2000, Uganda implemented the Uganda National Minimum Health Care Package (UNMHCP) in order to provide healthcare for the whole of its population. This was a basic package of essential healthcare services consisting of interventions against the most prevalent diseases in the country such as malaria, HIV/AIDS, diarrheal diseases, perinatal and maternal conditions to mention a few [12, 13]. The package was intended to be delivered by both public and private health facilities at all levels of healthcare provision. Lower-level private health facilities are often left behind during dissemination and training on new policies and guidelines and for this reason have lagged behind in implementing such policies [13–15]. In the rural areas of Uganda, private health facilities are more accessible to the population with in the recommended 5km radius compared to public health facilities. It is estimated that 50–60% of all sick individuals receive care from lower-level private health facilities, either as sole providers or as the first points of contact [16–19].

The capacity for lower-level private facilities to provide healthcare is often questioned and studies in Uganda have shown that compared to public facilities, private health facilities have less capacity in terms of equipment, infrastructure and adequate health workers' knowledge and skills [20, 21]. Varying data exist on quality of care for children in private facilities including their capacity to manage childhood illnesses. One study reported largely inappropriate management of common childhood illness at private sector drug shops in rural Uganda. Another study found no difference in the management for sick newborns in rural public and private facilities in eastern Uganda [22, 23]. A study in central Uganda reported great gaps between urban and rural private facilities with rural facilities having less trained health

personnel and less availability of some medicines [24]. We assessed the LLPHFs in Mbarara District in southwestern Uganda for diagnostics, vital medical supplies and infrastructure, and, facility health workers' knowledge required for management of common childhood illnesses including malaria, pneumonia, diarrhea, and, possible severe bacterial infections (PSBIs). Mbarara district is representative of the country; majority of the population is rural and the district has a homogenous distribution of health facilities with urban clinics and really rural clinics. A confidential inquiry into maternal and child deaths, carried out in two counties of Kashari and Rwampara in 2015 revealed child deaths exceeding the national figures. Majority of the health facilities are privately owned [7, 25].

## Methods

### Study design and study population

We conducted a cross-sectional study among low-level private health facilities and their health workers, in Mbarara District in southwestern Uganda between May and December 2019.

### Study setting

During the inception of this study Mbarara district consisted of 742 villages, 83 parishes, 16 sub-counties and 3 counties [26]. It had a mainly rural population estimated at 473,000 inhabitants. In 2019, the county of Rwampara became a district. In July 2020 Mbarara town was transformed into a city status by combining the sub counties of Kakoba, Kamukuzi, Nyamitanga, Biharwe, Kakiika and Nyakayojo. Fifty three percent of the households depend on subsistence farming for their livelihood [27]. The level of health facilities in the district is according to the Ugandan health system structure with facilities ranging from health centers (HC) to hospitals classified according to the population served, services provided, available infrastructure, and, staffing levels [28, 29] as illustrated in Table 1. Private health facilities are classified in a similar manner by the professional regulatory bodies, mainly for the purposes of determining the licensure fees.

Mbarara district has 7 hospitals, of which 6 are private. It also has 49 public health centers and 124 registered private health facilities. Of the 124 private facilities that were registered by regulatory authorities during the study period, 3 are HCIIIs, 3 are HCIIs, and the remaining 118 are private clinics and nursing homes with structures less than HCII level in reference to

**Table 1.  The health facility classification in Uganda.**

| Level of Health Unit | Target population | Services provided and structures |
|---|---|---|
| Village Health Teams (Health Centre I) | 1,000. | 1st contact for populations living in rural areas. Provides community-based preventive and health promotion services, community mobilization and referral of sick members to health facilities. No physical structures |
| Health Centre II | 5,000 | Parish level facility offering disease prevention, health promotion and outpatient curative health services for uncomplicated conditions, and immunization for children. |
| Health Centre III | 20,000 | Sub county level facility offering preventive, health promotion, outpatient curative, maternity, inpatient health services and laboratory services for malaria testing and TB microscopy |
| Health Centre IV | 100,000 | County level facility offering disease Preventive services, Health Promotion, Outpatient Curative, Maternity, inpatient Health Services, Emergency surgery and Blood transfusion and Laboratory services |
| General Hospital | 500,000 | District level facility. In addition to services offered at HC IV, offers general services and in-service training, consultation and research to community-based health care programs. |
| Regional Referral Hospital | 2,000,000 | In addition to services offered at the general hospital, offers specialist services, such as psychiatry, Ear, Nose and Throat (ENT), Ophthalmology, dentistry, intensive care, radiology, pathology, higher level surgical and medical services. |
| National Referral Hospital | 10,000,000 | Offers comprehensive specialist services and are involved in teaching and research. |

the ministry of health facility classification but higher than small drug shops [28, 29]. The facilities are registered by the councils to which the head or proprietor subscribes because of their qualifications. In this regard, 38 of the LLPHF are registered under Uganda Medical and Dental Practitioners' Council (UMDPC), 63 with the Allied Health Professionals' Council (AHPC) and 23 with Uganda nurses and Midwives Council (UNMC). The UMDPC registers qualified medical doctors and dental surgeons, the AHPC registers clinical officers, laboratory technologists/technicians and pharmacists while UNMC registers nurses and midwives.

For this study, we defined low-level private health facilities (LLPHFs) according to the Uganda Ministry of Health (MoH) health facility classification as those private facilities at HCIII and below, offering 'first-point-of-care" services to children [28, 29]. These included private for profit (PFP), private not for profit (PNFP), mission and non-governmental organization (NGO) facilities. In additionally, health facilities that were unregistered but acknowledged by local leaders as providing healthcare at a level higher than only selling drugs were also considered as LLPHFs. Pharmacies and retail drug shops were not included in the study. We included LLPHFs in all the three counties. To identify the registered LLPHFs, we utilized the lists of all registered health facilities from UMDPC, AHPC and UNMC. Unregistered LLPHFs were identified using local leadership at village level. For assessment of infrastructure, equipment, and medicines and supplies, the respondents were the supervising health workers in each LLPHF. For assessment of knowledge, we restricted responses to health workers who were directly involved in clinical care for sick children (e.g., medical consultations, and admissions), who were present on duty at the time of the facility visit.

### Data collection instruments

We used a checklist filled by a research assistant through interviewing health workers and observation. This allowed us to collect information on availability of trained staff, infrastructure, diagnostics, guidelines, medicines and health supplies. We used a self-administered knowledge tool to assess the knowledge of the health workers. The checklist was developed from the Integrated Management of Childhood Illnesses (IMCI) health facility assessment tool, the Uganda MoH Rapid Health Facility Assessment (R-HFA) tool and the Uganda Clinical Guidelines [30–32]. Any item available or missing at the facility was and noted on the checklist by the research assistant. The self-administered knowledge tool included a set of 10 vignettes each with 5 questions assessing the knowledge of the healthcare workers with regard to management of malaria, pneumonia, diarrhoea and possible serious bacterial infections as defined by IMCI [32]. The knowledge questions were developed with standard IMCI guidelines on management of the common childhood infections and the WHO IMCI distance learning course logbook [33]. The questions covered the key themes of clinical presentation, diagnosis, treatment, and indications for referral. JMA set the first draft of the questions which were then revised by the research consisting of academicians, pediatricians, epidemiologists and public health specialists. The questions were then piloted on 5 health workers from a public Health Centre III, to check for clarity.

### Study variables

Capacity was defined as the availability of vital and essential medicines, infrastructure, equipment, and, health workers with adequate qualifications and knowledge, for management of the common childhood infections at the LLPHF. According to the Uganda Clinical Guidelines (UCG), vital medicines are potentially life-saving, and their lack would cause harm and unfavourable outcomes. Essential medicines are those important medicines used to treat common illnesses but are not absolutely needed for example antipyretics [30]. The vital and essential

medicines included antimalarial drugs such as artemether/ lumefantrine, injectable artesunate, and oral and injectable quinine; zinc tablets, intravenous fluids and oral rehydration salts (ORS), and, vitamin A for diarrhoea; antibiotics for PSBIs and pneumonia including gentamicin, ampicillin, ceftriaxone, benzyl penicillin, amoxicillin, and, cotrimoxazole and paracetamol or ibuprofen for fever. We also looked out for availability of materials for mixing ORS, oxygen and blood for resuscitation and emergency transfusion. We intentionally looked for medicines which may be inappropriately used for managing some conditions, such as antimotility medicines for diarrhoea. In addition, we assessed availability of clinical guidelines and reference tools including the IMCI booklets, UCG, algorithm charts for malaria, pneumonia, diarrhoea, and, sick young infant. To assess the diagnostic ability, we collected data on availability of timers, tapes measures, stethoscopes, weighing scales, microscopes and rapid diagnostic tests (RDTs) for malaria and HIV. For infrastructure, we assessed for examination rooms with adequate lighting and examination couches and space for admission of patients, drug storage, and space for laboratory procedures. We assessed the availability of adequately trained staff, and whether the healthcare workers had received additional refresher training in IMCI guidelines in the past two years. In addition, we assessed the knowledge of the healthcare workers with regard to management of the common childhood illnesses (malaria, diarrhea, pneumonia and possible severe bacterial infection among young infants) using 10 vignettes in a self-administered questionnaire. Each vignette had 5 accompanying questions making a total of 50 questions. Likert scales were used to grade the responses to the questions on a 1–3 scale of strongly agree, neither agree nor disagree, and, strongly disagree (S2 File). Strongly agree and strongly disagree scored one point depending on whether it was correct or wrong. Neither agree nor disagree scored zero. All the correct answers to all questions carried equal weight so a simple summation was employed to get the total score out of 50 marks and percentage calculated [34]. A score below 50% was considered low, 50–70% as average and above 70% as optimal. For the population studied, the knowledge tool had a moderate Cronbach's coefficient alpha score of 0.63 as the measure of its reliability [35].

## Selection of the health facilities

We line listed 140 private health facilities in the study area using registers from professional regulatory bodies, the Mbarara District health Office and other identified by local leaders. Out of these, six were hospitals and 13 laboratories and were dropped from the list. We visited the remaining 121 facilities out of which four were permanently closed, six did not treat children, and, in one consent was not given by the owner. We collected data from all the remaining 110 health facilities. From each these facilities, one or two healthcare workers were purposively sampled for the knowledge assessment owing to their role in management of sick children. We used the same knowledge assessment tool for all the participants regardless of their professional qualification.

## Ethics approval and consent to participate

Ethical clearance to carry out the research was obtained from the Makerere University, School of Medicine Research and Ethics Committee (SOMREC) referenced #REC REF 2017–059 and the Uganda National Council for Science and Technology (UNCST) referenced SS 4903. To ensure anonymity providers at private clinics were de-identified by use of study numbers.

## Data management and analysis

We entered data into EpiData3. (EpiData, Odense, Denmark), and used Stata, version 13 (StataCorp, College Station, Texas, USA) for all the analyses. We conducted descriptive analysis;

reporting frequencies and percentages and used Chi square tests or 2-tailed Fisher's exact test to compare categorical outcomes between rural and urban facilities. We set statistical significance at $p$ value of 0.05. We considered LLPHFs to be rural if they were located outside the current Mbarara city divisions; facilities within the city divisions were considered urban LLPHFs. We stratified our analyses by location of the health facilities (rural vs urban) because of the previously observed discrepancies in the capacity between private health facilities elsewhere in the country [24], and to better inform possible targeted interventions.

We analyzed knowledge score by level of qualifications of the health workers (i.e., degree nurse/doctor, diploma/certificate nurse or midwife, diploma clinical officer, and, certificate nursing aids).

## Results

Data from 110 LLPHF were analyzed. We also analyzed data from 129 healthcare workers for knowledge.

### Baseline characteristics of the health facilities and healthcare workers

Of the studied LLPHFs, most (95%; n = 104) were of a level below HCII of the ministry of health facility classification, three were HCII and three at HCIII level. Forty-six (41.8%) of 110 facilities were located in urban areas while 64 (58.1%) were in rural areas. As shown in Table 2, majority of the health workers assessed for knowledge on management of common childhood illnesses, were certificate nurses/midwives and diploma clinical officers.

### Availability of medicines

The availability of antimalarial drugs, antibiotics, fluids and other medicines, by location status of the LLPHFs (rural areas versus urban areas) are presented in Table 3. Overall, oral antibiotics were available in more than 75% of the clinics. There was no statistically significant difference between availability of oral antibiotics in rural and urban facilities.

Most facilities stocked the important malarial medicines including Artemether/lumefantrine in 96% of the facilities and oral quinine in 78% of the facilities. Nearly all LLPHF had paracetamol and ibuprofen for fever management.

Most health facilities had zinc sulphate tablets and ORS for diarrhoea management however only 39% had vitamin A. The majority of the facilities had intravenous fluids and even some stocked 50% dextrose.

**Table 2. Qualification of the health workers assessed for knowledge in LLPHFs in Mbarara district.**

|  | N = 129 | |
| --- | --- | --- |
| Qualification of health worker | Number | Percentage |
| Nursing aid (Certificate) | 21 | 16.3 |
| Nurse/midwife (Certificate) | 67 | 52 |
| Nurse/midwife (Diploma) | 6 | 4.6 |
| Clinical officer (Diploma) | 28 | 21.7 |
| Nurse (Degree) | 1 | 0.8 |
| Medical doctor (Degree) | 6 | 4.6 |

**Table 3. Availability of medicines in LLPHFs in Mbarara district, Uganda by location of the health facility.**

| Variable | Total facilities, N = 110, n(%) | Urban (n = 46), n(%) | Rural (n = 64), n(%) | P value |
|---|---|---|---|---|
| **Antibiotics** | | | | |
| Oral amoxicillin | 107 (97) | 45 (98) | 62 (97) | 0.76 |
| Oral cotrimoxazole | 99 (90) | 42 (91) | 57 (89) | 0.70 |
| Injectable ceftriaxone | 95 (86) | 40 (87) | 55 (86) | 0.87 |
| Injectable gentamycin | 86 (78) | 36 (78) | 50 (78) | 0.99 |
| Oral ampicillin | 85 (77) | 37 (80) | 48 (75) | 0.50 |
| Injectable Benzylpenicillin | 85 (77) | 35 (76.1) | 50 (78.1) | 0.80 |
| Oral Penicillin V | 82 (75) | 33 (72) | 49 (77) | 0.57 |
| **Antimalarial** | | | | |
| Oral artemether/lumefantrine | 105 (96) | 42 (91) | 63 (98) | 0.16 |
| Oral sulphadoxine-pyrimethamine | 93 (85) | 41 (89) | 52 (81) | 0.26 |
| Oral quinine | 86 (78) | 26 (78) | 50 (78) | 0.99 |
| Injectable Quinine | 76 (69) | 31 (67) | 45 (70) | 0.74 |
| Injectable Artesunate | 55 (50) | 28 (61) | 27 (42) | 0.05 |
| Dihydroartemisinin-piperaquine | 41 (37) | 25 (54) | 16 (25) | 0.002 |
| **Medicines for diarrhoea management** | | | | |
| Zinc | 102 (93) | 45 (98) | 57 (89) | 0.14 |
| ORS | 103 (94) | 42 (91) | 61 (95) | 0.45 |
| Vitamin A | 43 (39) | 26 (57) | 17 (27) | 0.001 |
| Loperamide (Imodium) | 73 (66) | 28 (61) | 45 (70) | 0.30 |
| **Antipyretics** | | | | |
| Oral paracetamol | 107 (97) | 46 (100) | 61 (95) | 0.26 |
| Oral Ibuprofen | 104 (95) | 43 (94) | 61 (95) | 0.69 |
| **Fluids (intravenous)** | | | | |
| Normal saline | 89 (81) | 40 (87) | 49 (77) | 0.22 |
| 5% dextrose | 80 (73) | 32 (70) | 48 (75) | 0.53 |
| 50% dextrose | 69 (63) | 27 (59) | 42 (66) | 0.46 |
| Ringers lactate | 60 (55) | 28 (61) | 32 (50) | 0.26 |

## Availability of functional equipment and other medical supplies

Table 4 presents availability of other general supplies, laboratory supplies and reference tools for management of the common childhood infections, by the location of the LLPHF. The majority of the LLPHFs had accessories for intravenous fluid administration including giving sets and pediatric size cannulas. However, few facilities had basic materials used for assessing patients including pediatric weighing scales and MUAC tapes. Accessible means of transport for referrals was present in only 29 facilities.

Most facilities had materials and functional equipment for basic investigations such as microscopes and Rapid Diagnostic Tests (RDTs) for malaria and HIV. The distribution of laboratory supplies and equipment for management of common childhood infections was not statistically different between rural and urban facilities.

Apart from the Uganda clinical guidelines, only a few health facilities had charts and algorithms including the IMCI chart booklet used as guides to ensure standardized management of common infections in children.

**Table 4. Availability of medical supplies and equipment in LLPHFs in Mbarara district, Uganda by location of the facilities.**

| Variable | Total facilities, N = 110, n (%) | Urban (n = 46), n (%) | Rural (n = 64), n (%) | P value |
|---|---|---|---|---|
| **General supplies** | | | | |
| Cannulas (for children | 96 (87) | 40 (87) | 56 (88) | 0.93 |
| Fluid giving sets | 89 (81) | 37 (80) | 52 (81) | 0.92 |
| BP machine | 89 (81) | 38 (83) | 51 (80) | 0.68 |
| Stethoscope | 85 (77) | 36 (78) | 49 (77) | 0.22 |
| Clock/ timer | 63 (57) | 31 (67) | 32 (50) | 0.07 |
| Baby weighing scale | 21 (19) | 9 (20) | 12 (19) | 0.92 |
| Adult weighing scale | 55 (50) | 28 (61) | 27 (42) | 0.03 |
| Supplies to mix ORS | 39 (36) | 14 (30) | 25 (39) | 0.35 |
| MUAC tape | 31 (28) | 21 (46) | 10 (16) | 0.001 |
| Tape measure | 31 (28) | 20 (44) | 11 (17) | 0.003 |
| Accessible transportation for patients referrals | 29 (26) | 14 (30) | 15 (23) | 0.41 |
| Oxygen source | 6 (5.5) | 2 (4.4) | 4 (6.3) | 0.67 |
| **Laboratory** | | | | |
| Malaria RDT | 95 (86.) | 40 (87) | 55 (86) | 0.88 |
| Microscope | 95 (86.) | 40 (87) | 55 (86) | 0.88 |
| HIV RDT | 91 (83) | 38 (83) | 53 (83) | 0.98 |
| Glucometer | 57 (52) | 28 (61) | 29 (45) | 0.11 |
| **Reference tools** | | | | |
| Uganda clinical guidelines | 74 (67) | 29 (63) | 45 (70) | 0.36 |
| Malaria algorithm chart | 18 (16) | 7 (15) | 11 (17) | 0.78 |
| Pneumonia algorithm chart | 14 (13) | 7 (15) | 7 (11) | 0.51 |
| Sick young infant algorithm chart | 14 (13) | 7 (15) | 7 (11) | 0.51 |
| Diarrhea algorithm chart | 12 (11) | 6 (13) | 6 (9.4) | 0.54 |
| IMCI booklet | 8 (7.3) | 5 (11) | 3 (4.7) | 0.28 |

## Availability of health workers trained in IMCI, and, infrastructure

Only a few of the health facilities had health workers who had ever trained in IMCI as is shown in Table 5. This finding was similar for both rural and urban health facilities.

More than half of the facilities had an examination room with appropriate lighting, and patients' admission rooms. But less than half of the facilities had a laboratory.

## Knowledge on management of common childhood illnesses

The mean percentage knowledge score was 62% (SD±14) for the 129 respondents. Majority of the health workers (n = 90; 70%), had average knowledge, 23 (18%) had low knowledge, and, 16 (12%) had optimal knowledge on management of common childhood illnesses. After stratifying level of knowledge by qualification, all of the degree nurses or doctors had average or optimal scores of knowledge, as shown in Fig 1. The differences in knowledge levels across the different professions was not statistically significant ($P = 0.09$).

## Discussion

We assessed the capacity of LLPHFs to manage common childhood illnesses including malaria, pneumonia, diarrhea, and, PSBIs in Mbarara District, southwestern Uganda. We found that most of the LLHFs in rural and urban settings had first-line medicines for treatment of the common childhood illnesses; but lacked clinical guidelines and reference tools,

**Table 5. Availability of health workers trained in IMCI, and, infrastructure in LLPHFs in Mbarara district, Uganda by location status of the facilities.**

| Variable | Total facilities, N = 110, n (%) | Urban (n = 46), n (%) | Rural (n = 64), n (%) | P value |
|---|---|---|---|---|
| **Human resource** | | | | |
| **IMCI trained medical doctor** | | | | 0.6 |
| None | 96 (87) | 37 (80) | 59 (92) | |
| One | 10 (9) | 6 (13) | 4 (6) | |
| Two or more | 4 (3.6) | 3 (6.5) | 1 (2) | |
| **IMCI trained nurses/midwives** | | | | 0.28 |
| None | 68 (64) | 26 (57) | 42 (69) | |
| One | 26 (24) | 12 (26) | 14 (23) | |
| Two or more | 13 (12) | 8 (17) | 5 (8) | |
| **IMCI trained nursing assistants** | | | | 0.38 |
| None | 107 (97) | 44 (96) | 63 (98) | |
| One | 3 (3) | 2 (4) | 1 (2) | |
| **IMCI trained clinical officers** | | | | 0.30 |
| None | 90 (82) | 35 (76) | 55 (86) | |
| One | 16 (15) | 8 (17) | 8 (13) | |
| Two or more | 4 (4) | 3 (7) | 1 (2) | |
| **Infrastructure** | | | | |
| Examination room with appropriate lighting | 69 (63) | 32 (70) | 44 (69) | 0.93 |
| Laboratory room | 50 (46) | 24 (52) | 26 (41) | 0.48 |
| Injection room | 72 (66) | 30 (65) | 42 (66) | 0.90 |
| Drug store | 65 (59) | 29 (63) | 36 (56) | 0.56 |
| Drug shelves | 90 (82) | 35 (76) | 55 (86) | 0.19 |
| Waste pit | 64 (58) | 24 (52) | 40 (63) | 0.28 |
| Inpatient room | 76 (69) | 31 (67) | 45 (70) | 0.74 |
| Admission beds, median (IQR) | 2 (0–4) | 2 (0–4) | 2 (0–4) | 0.39 |

including IMCI booklets and algorithm charts for the management of the childhood infections. Some facilities had oxygen which is not found at facilities of same level in public sector. Further, the majority of the health workers in the facilities scored averagely to poorly in knowledge of managing the common childhood illnesses.

## Medicines, medical supplies and diagnostics for malaria

This study found that the recommended first-line antimalarial drug artemether/lumefantrine [30], microscopes and rapid diagnostic tests (RDTs) were available in more than 85% of the LLPHFs. These findings were comparable to a recent study that found high stocks of artemether/lumefantrine in the private health facilities [21]. Our findings however contradict findings of a previous study in central Uganda that reported a lower proportion of private facilities with diagnostic facilities including RDTs and microscopes [24] but it is worth mentioning that the study in central Uganda included drug shops and pharmacies that may not stock diagnostics. The findings in our study are reassuring with regard to antimalarial medicines and diagnostics availability as they seem to suggest that these facilities have the capacity to properly diagnose and treat uncomplicated cases of malaria. In reality, this would only be possible if the other conditions for provision of quality care are in operation, including health workers with adequate knowledge on management of malaria. A recent article by our research team described inappropriate care for malaria and other common pediatric infections, implying that it requires more than availability of medicines and diagnostics [36]. The fact that the first

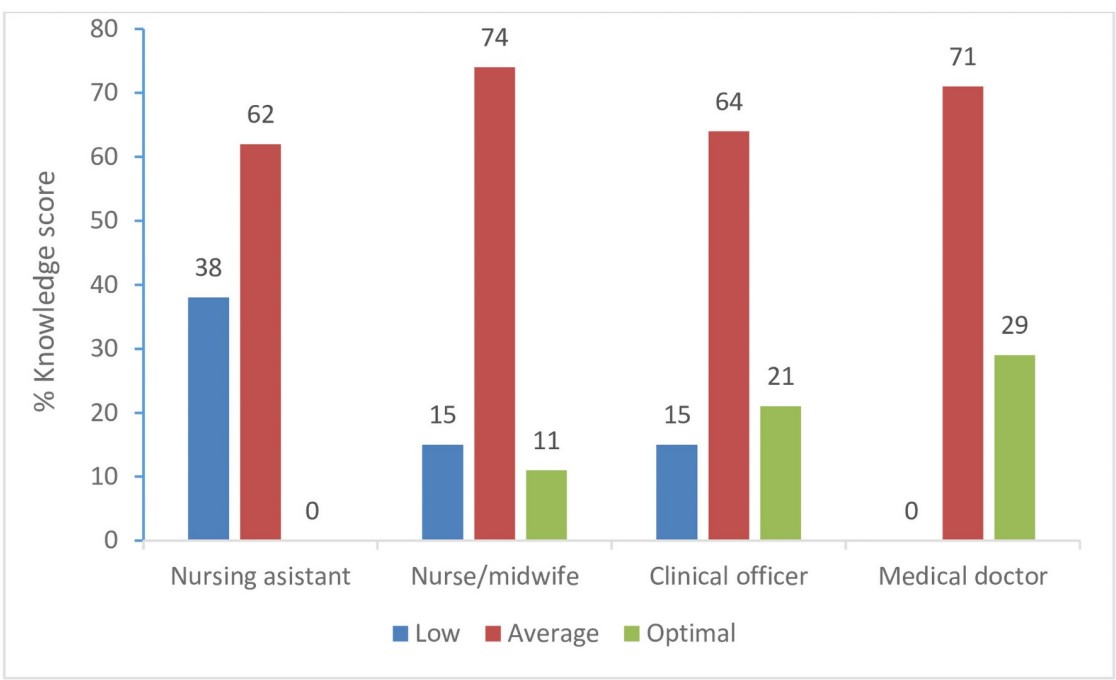

**Fig 1. Level of knowledge on management of common childhood illnesses by health workers in LLPHFs in Mbarara district, Uganda by the qualification of the healthcare workers.**

line antimalarial medicine for severe malaria, injectable artesunate, was not available in half of the facilities, may imply that children with complicated malaria may not be getting appropriate pre-referral treatment.

## Antibiotics and oxygen for managing pneumonia and PSBIs

We found that the essential antibiotics used in treatment of children with pneumonia and possible severe bacterial infections were available in both rural and urban facilities, consistent with previous findings in Uganda [21, 24]. Available essential antibiotics are only useful when used in a responsibly. However, studies from countries at all income levels have shown that antibiotics are often used irrationally which contributes unnecessarily to antibiotic resistance [37–39]. The same is also true for antimalarials. It is important to note that very few facilities had oxygen sources implying that very few would be able to resuscitate patients presenting in critical conditions before referring them. This finding is consistent with that of other studies in Uganda and other sub-Sahara African countries which describe inadequate oxygen availability in both public and privately owned health facilities at hospital and levels [40–42]. This contributes to poor outcome in treatment of pneumonia or other severe illnesses.

## Medicines and supplies for managing diarrhoea

Nearly all LLPHFs (>90%) facilities in rural and urban had stocks of zinc sulphate tablets and ORS for management of diarrhea, contrary to previous findings in central Uganda in 2014 [24]. This observed increase in availability of zinc tablets and ORS may be attributed to interventions that aimed to improve access to these drugs through price reductions, and generation of demand through a variety of platforms including trainings, and use of community groups

and Village Health Teams (VHTs) to promote ORS/Zinc in the recent past [43]. However, the supplies for demonstration of how to prepare and administer ORS were available in only 36% of the facilities. This may imply that most health workers do not routinely demonstrate how to prepare ORS to caretakers of children with diarrhea and yet correct use of ORS is dependent on its correct preparation and administration [44]. Only 39% of facilities had vitamin A in stock and the situation was worse in rural facilities. This is a reflection of the already known low status of vitamin A supplementation for children below 5 years in the country, which needs to be improved since vitamin A supplementation is important for child survival and management of persistent diarrhoea [45]. It is important to note that more than half of the facilities stocked loperamide hydrochloride, an antimotility drug. This could be a proxy indicator that it is used to manage children with diarrhoea and yet it is contraindicated in management of diarrheal diseases in children of ages below 3 years because it has unfavorable side effects [46, 47].

## Other tools for examining children

The survey findings of low availability of basic equipment used for assessing children including weighing scales, MUAC tapes, and timers may imply incomplete assessment of children.. A study carried out in several countries including Uganda found that assessments for children with respiratory symptoms was often incomplete with health facilities in Uganda rarely counting the respiratory rates [48]. In these facilities with inadequate equipment for assessing sick children such as weighing scales cases of malnutrition which may require special treatment among the children presenting with the common childhood infections can be missed and thus inappropriately treated. It is likely that the healthcare workers in these facilities prescribe drugs for common childhood illnesses based on age rather than weight as has been highlighted in earlier studies in other sub-Saharan African countries [49, 50], which may result in inappropriate dosing. Therefore, efforts should be made by the licensing bodies and the Ministry of Health to ensure that these facilities have the appropriate equipment in place.

## Knowledge and training of the healthcare workers

More than 70% of the facilities lacked health workers trained in IMCI and the IMCI booklets and algorithm charts for management of common childhood illnesses were also rare in these LLPHFs. This is comparable to another study carried out in Nigeria [51] and may explain the lack of good knowledge on management of these illnesses, exhibited in this survey especially among the lower professional cadres. This calls for training of health workers in LLPHF in IMCI and availing them with guidelines. A study in Afghanistan showed improved knowledge and skills of primary care health workers when they received regular training in IMCI [52].

## Limitations

This study has some limitations.. The tool used to assess the knowledge had alternatives from which the respondent would pick the best answers; therefore, a possibility of getting correct answers through guessing. In addition the tool had never been validated and had a moderate reliability Cronbach's coefficient alpha score of 0.63. Nevertheless the fact different cadres of health workers scored different marks, albeit no statistical difference, supports its validity. As the next steps, we need to refine this scale so that it can be used in similar study settings. These limitations notwithstanding, our survey has provided useful information on the operational capacity of LLPHFs to manage malaria, pneumonia, diarrhoea and possible serious bacterial infection among children in Mbarara District and settings with similar operational capacities. The findings serve to inform policies on child health interventions in the country.

## Conclusion

This survey revealed availability of common first-line medicines for treatment of pneumonia, diarrhea, and possible severe bacterial infections in most of the LLPHFs. However, most facilities lacked appropriate job aides, and some health workers who were inadequately trained on IMCI guidelines and lacked good knowledge on management of the illnesses. There is need for knowledge raising interventions in both rural and urban LLPHFs including refresher trainings on IMCI guidelines and availing of job aides to improve the knowledge of healthcare workers in these facilities. This may translate into better management of common childhood illnesses in the region.

## Supporting information

**S1 File. De-identified dataset.** Health facility assessment tool.
(DTA)

**S2 File. Knowledge testing tool.**
(PDF)

## Acknowledgments

We acknowledge all the health workers and policy makers who accepted to be interviewed and the owners of the private facilities who allowed the interviews to take place in their premises. RAs Bob Beinomigisha, Ruth Mbabazi, Ahabwe Chris, Patricia Tushemereirwe, Beatrice Katusiime, Collins Ogola and Edgar Atwine for data collection.

## Author Contributions

**Conceptualization:** Juliet Mwanga-Amumpaire, Grace Ndeezi, Karin Källander, Celestino Obua, Cecilia Stålsby Lundborg, Joan Nakayaga Kalyango, Tobias Alfvén.

**Data curation:** Juliet Mwanga-Amumpaire, Richard Migisha, Juvenal Nkeramahame.

**Formal analysis:** Juliet Mwanga-Amumpaire, Richard Migisha, Joan Nakayaga Kalyango.

**Funding acquisition:** Celestino Obua.

**Investigation:** Juliet Mwanga-Amumpaire.

**Methodology:** Juliet Mwanga-Amumpaire, Grace Ndeezi, Karin Källander, Celestino Obua, Cecilia Stålsby Lundborg, Tobias Alfvén.

**Project administration:** Juliet Mwanga-Amumpaire.

**Supervision:** Juliet Mwanga-Amumpaire, Grace Ndeezi, Karin Källander, Celestino Obua, Cecilia Stålsby Lundborg, Joan Nakayaga Kalyango, Tobias Alfvén.

**Validation:** Grace Ndeezi, Karin Källander, Cecilia Stålsby Lundborg, Joan Nakayaga Kalyango, Tobias Alfvén.

**Writing – original draft:** Juliet Mwanga-Amumpaire, Richard Migisha, Juvenal Nkeramahame, Joan Nakayaga Kalyango.

**Writing – review & editing:** Juliet Mwanga-Amumpaire, Grace Ndeezi, Karin Källander, Celestino Obua, Richard Migisha, Juvenal Nkeramahame, Cecilia Stålsby Lundborg, Joan Nakayaga Kalyango, Tobias Alfvén.

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
