## [Decision Letter · Decision Letter 0]

21 Jul 2021

PONE-D-21-18488

Capacity to provide care for common childhood infections at low-level private health facilities in Western, Uganda

PLOS ONE

Dear Dr. Mwanga-Amumpaire,

Thank you for submitting your manuscript to PLOS ONE. After careful consideration, we feel that it has merit but does not fully meet PLOS ONE’s publication criteria as it currently stands. Therefore, we invite you to submit a revised version of the manuscript that addresses the points raised during the review process.

Editor comments:

After review of the manuscript, I have no concerns about the quality or clarity of the writing. The majority of the reviewers felt the manuscript was well written. When assessing the stylistic suggestions of Reviewer #3, the authors should use their discretion on whether to modify the language. I consider these edits to be non-essential. In addition, I leave it up to the authors whether they would like to consider major structural changes to the organization and presentation of the results within the manuscript. However, there are some concerns noted about redundancy with constructive suggestions outlined by Reviewer #1. 

One area that must be addressed is the question raised related to the sampling of the hospitals. I recommend the authors follow the suggestions outlined by by Reviewer #1 with additional consideration to the comments by Reviewer #3, if applicable. 

Please address all relevant comments in the revised manuscript and address each comment in a detailed point-by-point response. In the event the authors do not feel any edits are required, please indicate so and justify it in the response to reviewers.

We look forward to receiving your revised manuscript.

Kind regards,

Andrea L. Conroy, PhD

Academic Editor

PLOS ONE

Journal Requirements:

4. Please ensure that you refer to Figure 1 in your text as, if accepted, production will need this reference to link the reader to the figure.

5. Please upload a copy of Figure 2, to which you refer in your text on page 16. If the figure is no longer to be included as part of the submission please remove all reference to it within the text.

Reviewers' comments:

Reviewer's Responses to Questions

**Comments to the Author**

1. Is the manuscript technically sound, and do the data support the conclusions?

Reviewer #1: Yes

Reviewer #2: Yes

Reviewer #3: No

2. Has the statistical analysis been performed appropriately and rigorously? 

Reviewer #1: Yes

Reviewer #2: Yes

Reviewer #3: Yes

3. Have the authors made all data underlying the findings in their manuscript fully available?

Reviewer #1: Yes

Reviewer #2: Yes

Reviewer #3: Yes

4. Is the manuscript presented in an intelligible fashion and written in standard English?

Reviewer #1: Yes

Reviewer #2: Yes

Reviewer #3: No

5. Review Comments to the Author

Reviewer #1: The authors are to be congratulated on this practical study. They conducted a census of quality of care for pediatric illnesses in a large number (n=110) of health facilities in Western Uganda. This information is helpful in highlighting gaps in clinical care. I would be in support of publication of these data, recognizing that they are quite specific to the locale (not generalizable). Nonetheless, this setting is not unique within Uganda and the broader sub-Saharan African healthcare landscape. It is important to highlight ongoing gaps in clinical care; there is a paucity of such data in the published literature.

Major comments:

1. “sample” versus total population of health facilities in Mbarara District

P9 line 177: “We randomly selected the LLPHFs from the list of all identified LLPHFs by the licensing bodies 178 and local leadership”

and P10 line 200: “data were not collected from 30 facilities”

My understanding of the selection of included facilities is as follows: There were a total of 140 facilities (124 registered and 16 identified through local leaders), of which 30 did not provide data because they met exclusion criteria such as hospital, no pediatric services, lab only, etc, leaving 110 facilities.

This appears to be the entire set of eligible facilities in the Mbarara District (please clarify if there were more than 140). If that is the case, this is not a sample, but the entire group of facilities (“population”). If I have understood this correctly, then the sample size calculation is not germane, since all sites were included, it is not a sample. Furthermore, the methods state that a random sample was selected, however, this is not a random sample but the entire set of eligible facilities.

If I have understood correctly, please delete the sample size calculation and the statement about a random sample, and simply state that all registered and unregistered facilities in the District were included, with exclusions for hospital, no pediatric services, lab only, etc.

If this is not correct, then the site selection needs to be explained more clearly.

2. Knowledge scale not validated

P9 lines 168: “A score below 50% was considered low, 50-70% as average and above 70% as optimal.”

This is quite arbitrary. It is not clear what the implications (for quality of nursing care or clinical care) would be for different levels of knowledge. The fact that the scale is not validated and has not been associated with markers of quality of care or patient outcomes needs to be mentioned in the limitations section. What we have here is a scale (number) without an interpretation.

P9 lines 169: “Cronbach’s coefficient alpha score of 0.63.” Reliability is moderate. This knowledge score requires further validation and probably refinement of the scenarios, modification or even deletion of some questions, depending on their performance in contributing to a unified construct. Please list this as a limitation in the discussion (scale not previously validated). Different scores among health workers in different cadres was supportive of the validity of the scale, although this did not reach statistical significance.

3. Most of the text in Results section is redundant (duplicates Tables).

P11-13: The text of the Results duplicates information found in the Tables. This is not necessary and is redundant. Recommend major reduction of the text and just simply referring to the table, without repetition of numbers that can be found in the Table.

Minor comments:

P7 line 119 “in-charges” is a colloquial term. Prefer “charge nurse” or other formal title.

P8 line 148 “gentamycin” spelling error. correct spelling: “gentamicin”

P9 lines 165-7: “Likert scales were used to grade the responses to the questions on a 1-3 scale of strongly agree, neither agree nor disagree, and, strongly disagree (Appendix 1).”

This is unusual for a knowledge items which can be scored as correct or incorrect. Likert scales for gradation of agreement seem less applicable here (better suited for "attitudes"). How was a response of “neither agree nor disagree” scored? (I presume incorrect).

P11 line 213: “rural areas vis-à-vis urban areas” should be “versus”

Tables 3,4: A study with n=110 cannot measure proportions with such precision (3 significant figures are reported, implying a precision beyond that which is possible with this sample size, +/- 5%, approximately). Please round all percentages to 2 significant figures (as in Table 5 – this one is great!). The P-values are also excessively precisely reported with 3 significant figures – recommend rounding to 2 sig figs.

Figure. Regarding the number on top of each error bar: the number of participants (or n/N) would be more helpful than providing the percentage (which can be read off the y-axis). Providing the number of participants also helps to understand the precision with which this proportion is estimated. Could consider illustrating this on the graph with error bars, representing the binomial 95% confidence interval on the proportion, for example.

The literature review for the Discussion is acceptable, but I am aware of some studies (e.g., on oxygen availability in Uganda and elsewhere) that would serve as good comparator studies, and might be considered for inclusion in the discussion:

• Otiangala D, Agai NO, Olayo B, Adudans S, Ng CH, Calderon R, et al. Oxygen insecurity and mortality in resource-constrained healthcare facilities in rural Kenya. Pediatr Pulmonol. 2020;55:1043–9.

• Nabwire J, Namasopo S, Hawkes M. Oxygen availability and nursing capacity for oxygen therapy in Ugandan paediatric wards. J Trop Pediatr. 2018;64(2):97–103.

• Belle J, Cohen H, Shindo N, Lim M, Velazquez-Berumen A, Ndihokubwayo J, et al. Influenza preparedness in low-resource settings: a look at oxygen delivery in 12 African countries. J Infect Dev Ctries. 2010b;4(7):419–24.

Reviewer #2: General comment

Well-written manuscript on a topical issue affecting health care delivery in Uganda. The availability of essential medicines in the low-level health facilities is reassuring .The purely descriptive approach however adds little to the knowledge as to why there was low knowledge among the health workers. Authors need to clarify several statements especially in the results section.

Abstract

The abstract is well written and conclusions are drawn from the data presented.

Introduction and background

‘69 The package was intended to be delivered by both public and

private health facilities as it was developed to cover all levels of health care even though lower

70 level facilities lagged behind [13-15]. Sentences need to be re written for clarity. Package meant for both public and private facilities, also developed for all levels Lower levels have lagged behind’ – In doing what? Since when? At this point, a definition of levels of health care facilities in Uganda would help the readership under stand lower level facilities are.

7”9 For example, one study reported largely inappropriate management of common childhood 80illness at private sector drug shops in rural Uganda, while another study found no difference in 81the management for sick newborns in rural public and private facilities in eastern Uganda 82[22, 23]. Another study in central Uganda reported great gaps between urban and rural private 83facilities with rural facilities having less trained health personnel and less availability of some medicines”

How comparable are drug shops and say lower level clinic?

Justification for doing the study – varying information about the capacity to care for children in private and public health care facilities- The authors need to clarify the justification for a study in lower level facilities in Mbarara district. Could a multi center study – multi region study yielded more meaningful and more generalizable information? How different is Mbarara district from the districts in central Uganda where earlier studies were done?

Methods

Assessment tools for knowledge – Could it be possible that some of the respondents got the answers correct purely by chance or guess work? How did the authors control for this? Would perhaps self-rating scales or focus group discussions or reflective practice sessions have added value to the assessment tool?

Results

‘Data from 110 out of the 140 private listed facilities, 124 of which were registered by regulatory

199 bodies and 16 identified by local leaders, were analyzed-

This statement is confusing, how many health facilities were studied? How many had complete data? How many facilities had their data analyzed? ‘

Out of these, data were not collected from 30 facilities; six of them were hospitals, four facilities were permanently closed, 13 facilities only offered laboratory services, and seven that did not treat children.

Out of which ones? The 110, 140 or 124?

The majority of the health facilities (95%; n=106) of 110

203 LLPHFs were of a level below HCII of the ministry of health facility classification; three were HCII and three at HCIII level. The numbers do not add up, if n=110, and 106 were below HCII, and then the balance should be 4, and not 6 health centers as shown by the authors numbers.

Overall, but more stocked by urban (54%) compared to rural facilities 222 (25%), P=0.002. Nearly all LLPHF had paracetamol (97%) and ibuprofen (95%) for fever management-

Would be nice to add the numbers (n/N) to add meaning to the percentages.

Availability of medical equipment

Impressive results – Were the authors looking out for availability or the use or functionality of these equipment at the health facilities?

Reviewer #3: General comment

This is an important paper; reflecting on the preparedness and readiness of private health facilities in Mbarara to provide care to children with common childhood illnesses. The role of the private sector in providing health services is a topical area, with their exact role in complementing the public health service delivery system ill defined. The findings of the paper are undoubtedly a valuable contribution to the debate. However, despite its importance the paper is poorly written, methods lacking, results shallow and poorly presented, and the discussion lacks in depth and is premised on too many assumptions rather than actual study findings. Provided below are comments.

Specific comments

Title: The title could be more specific; specify the location where the study was done; Mbarara district. 'Western Uganda' is too wide.

Key words (Page 3 line 48): should appear on the title page, not after the abstract

Abstract (Page 2)

The presented results should match those in the main body of the manuscript and in the tables. They seem to be discrepancies

INTRODUCTION

General comments

― For purpose of clarity, the authors should re-write the entire introduction paying attention to spelling, punctuation, and grammar.

― The justification of the study is not well captured in the introduction

Specific comments

Page 3, line 52: The first paragraph of the introduction starts with a complex sentence. The sentence is: 1) too long, 2) comprised of too many clauses, and 3) starts with a subordinate clause, the second clause being the main idea. For purposes of scientific writing and clarity it may be best to start the sentence with the main clause. Use of a subordinating conjunction ‘As’ maybe grammatically correct for a sentence, but is not the best way to start a paragraph.

Page 3, line 55: The 2nd sentence, like the preceding one starts with ‘As’ Using ‘As’ repeatedly (appears multiple times in the introduction) makes the reading repetitive.

― Revise the 1st and 2nd sentences of the first paragraph. Good to start with clarity and to the point.

Page 3, line 56: The 4nd sentence starting “In 2019, Sub Saharan Africa was reported to be responsible….” is grammatically incorrect. SSA cannot be responsible for death, but may have accounted for….

Page 4 line 65: The sentence starts with a conjunction ‘in order’ again. The sentence should start with the main clause, thus: In 2000, Uganda implemented……… Additionally, in the same sentence (line 67) the words ‘disease conditions’ sound repetitive, either word would have sufficed

Page 4 line 70: The sentence starting “In the rural areas of Uganda…” implies private facilities are more accessible within the recommended 5km. How about outside the 5km?

The authors use the words services and care interchangeably, there is need to be consistent with choice of words expressing the same point

METHODS

General comments

― For clarity, include more specific sub-sections avoid merging different subjects in the same sub-title

― Include the following subtitles: study design, identification and selection of facilities,

Page 5 line 91: Separate the sub titles, study setting and study population. Comment on the level of development in Mbarara and the social economic status of the population

Table 1 is not necessary, a summary of the structure can be provided in text.

Page 5 line 94: The sentence starting “The health facilities in the district follow the Ugandan…….” Did the facilities really follow the Uganda health system? Use of the word follow as a verb is inappropriate.

With respect to health care system at the district level, mention: 1) decentralized health system, and 2) the HSD a functional division of the district health system, equivalent to a constituency/county. Service delivery in the HSD is provided at outpatient facilities tiered at three levels 2 (parish), level 3 (sub county), level 4 (county). This system is applicable to public health care system, what about private sector? Is it the same or different? Please elaborate the structure (types and levels) of the private health care providers.

Page 6, lines 102-103…the statement, “….the remaining 118 are private clinics and nursing homes which are below HCII level in reference to the MOH guidelines.” This means these facilities were level 1-health centers. The facilities are registered by councils; during registration the facility type and level should have been defined. Were these details available, Please clarify?

Page 6, line 110: the definition of LLPHFs is vague. First, use of the word 'equivalence' is redundant, as some of the facilities were already defined as either level II or III facilities. Second, for those that didn’t have a level how was equivalence to level II/III determined? What criterion was used?

Page 6, line 116: How were drug shops distinguished from other private providers. It should be noted that the potential for overlap between drug shops and private clinics is high. It’s possible that drug shops were captured as clinics. Please elaborate on how this was minimized.

Page 7, line 125: Data collection instruments

― The write up on data collection instruments is disorganized and sloppy. For clarity, list the questionnaires and for each state: 1 the purpose of the questionnaire, 2) who administered the questionnaire, and 3) who the respondent was.

― It’s stated (Page 7 LINE 125) “….self administered questionnaire to assess capacity of LLPHF. This is misleading; later on (page 7 line130 -132) it appears that self administered questionnaire were used to specifically assess KAP

― Line 130 The statement “10 vignettes with 50 questions.” is ambigous. Does it mean each vignettes was comprised of 50 questions, or all added up to 50 questions. Would be best to state how many questions per vignettes. While describing the vignettes you in-appropriately referred to them as knowledge questionnaires, which is a bit confusing. Were the questions focused on specific diseases, if yes, please indicate them

Page 8-9: Study variables

― The WHO categorizes outcome measures for health facility assessments in terms of ‘service availability’ and ‘service readiness.’

― The authors need to ascertain if this was a 'service availability' and ‘service readiness’ survey.

― Use of the term ‘capacity’ undermines the scope of work done.

― The term capacity of the health facility should not be mixed up with health workers qualification and knowledge

― Page 8, line 144-145: How does lack of vital medicines cause side effects?

― Page 8, line 151-152: Facilities should have been assessed based on expected capacity. Assessing facility performance against standards above their ability is meaningless. For example, do we expect lower level facilities to be providing Oxygen and blood transfusion service? Probably not, and if they did, that would translate to be a problem.

― Page 9 Line 164-165: the description of the vignettes is more clear than was on Page 7, line 140

― Page 9 Line 167-165: All questions carried equal weight…should be written as 'the correct answers to all questions carried equal weight'

― Page 9 Line 168-169: What was the basis of the performance cut offs.

Page 9 line 165-168: Can the authors provide more detail on how they analyzed (scored and summarized) data collected using the likert scales. The scales comprised of 3 options, agree, neither agree or disagree and disagree. How were right and wrong answers handled? and what was the overall distribution of responses based on the 3 options.

Page 9: Sample size and sampling

― Create an independent sub-section for sampling

― What sampling procedure was used to select facilities

― Why did you sample one or two health workers; was a representative sample obtained

Results

Page 10 Lines 200-203. The first and second sentences are poorly written and confusing. The order of sequencing the events leading to enrollment of facilities that were studied is not logical. A study profile highlighting how facilities were selected from the entire pool would be helpful.

Page 10 Lines 204: The sentence starting “The majority of the health facilities (95%; n=10) of 110…. ” is poorly written. Start with the overall state. Thus: Of the studied LLPHFs, most (106; 95%) were below level II………….

Page 10 Line 204-206: Of 110 facilities, 106 were below level II, three were level II, and three were level III giving a total 112 facilities greater than the number of facilities (110) enrolled. Where did the extra two come from? Correct this error

Page 10 Line 204-206: Of 110 facilities. 106 were below level II, three were level II, and three were level III giving a total 112 facilities greater than the overall number of facilities (110) enrolled. Where did the extra two come from? Correct this error

Page 10 Line 206-207: Urban and rural…how were these defined and determined? Mbarara city is urban; its surroundings are peri-urban. Therefore, facilities in peri-urban areas could have been captured as rural. Kindly clarify

Include a sub-title ‘Baseline characteristic of health facilities and health care workers’ Baseline results of facility and health care workers should be reported under this section. Other characteristics to consider in this section include:

― Health care worker age, gender, and year of training.

― Facility ownership and or registration status

Replace table 2 with a table of baseline characteristics combining both facility and health care worker characteristics.

Page 11 Availability of medicines

The word ‘overall’ is used repeatedly; it is best placed at the start of the sentence.

Page 11 Line 217: replace pneumonia with ‘non-severe pneumonia’

Page 11 Line 218: include the results for availability of antibiotics in rural and urban facilities.

Page 11 Line 218: include the results for availability of antibiotics in rural and urban facilities.

Page 11 Line 219: Parenteral ampicillin is referred to in this sentence; however, it’s not captured in Table 3. Additionally, why are the results for parenteral antibiotics (ampicillin, penicillin, gentamycin) combined? They should be presented independently.

Page 12 Line 231: Table 3. In addition to urbanization level, results should be stratified by facility level and registration status. Important differences could arise within these sub-groups

Page 11 Line 222: The result presented for Artemether/Lumefantrine is different from that in Table 3. Same problem applies for the results of all others medicines presented in the text; they don’t match the results presented in Table 3. Please correct for consistency.

For comparison of % across urban vs. rural facilities, can the absolute difference be included with the 95%CI around the difference.

Page 13 Line 234 Availability of equipment and other medical supplies

The proportions presented in this section, do not match those in Table 4; they should be the same.

Page 14 line 251: Table 4. Can all the p-values be re-calculated? Some seem to be wrong. For example, adult weighting scale; the significance of difference between urban and rural proportion returns a p-value of 0.053 and not 0.032

Page 14 line 251: Table 4. Like table 3, results should be stratified by facility level and registration status. Important differences could exist between these sub-groups

Page 14 line 55: Infrastructure and availability of trained human resource

The title could be made more specific ‘Availability of infrastructure and healthcare workers trained in IMCI’

The order of presentation of results should match the order in the title. Therefore start with infrastructure

The order in which the results are presented in the sub-section does not match the order of presentation in Table 5.

Page 14 Line 256: The first sentence states, “Of 110 facilities, 14 (13%) had at least one doctor trained on IMCI…….as shown in Table 5. However, in Table 5, the category for at least one trained doctor does not exist. Additionally, for nursing assistants the catagorey 'Two or more' is missing

Page 15 Line 261: Table 5, What is the importance of the order of ranking for the number of health care workers: “None,” “One,” and “Two or more.” Does being trained two or more times matter more than being trained once? What would have been more important is the duration since last training. Additionally, why only focus on IMCI training, how about other trainings, why were these not considered?

Page 15 Line 261: Table 5, for the characteristic ‘Examination room with appropriate lighting’ the value for Urban (32) and rural (44) don’t add up to the total (69)

Page 15 Line 264: First sentence “….about 63%....” is sloppy writing, ‘about’ suggests uncertainty. In previous sections results are presented as an absolute number with % in parenthesis. In this section, results are presented a % only. Need to be consistent.

Page 16 Line 271: "Upon stratification by health worker qualification......" can the knowledge results be presented positively, i.e. those who passed rather than those who did not fail.

Page 16 Line 273: Degree nurses are not included in the Figure.

Page 16 Line 273: Figure 2 should be Figure 1

Discussion

Page 16 Line 284: Lack of oxygen should not be presented as a gap with respect to these LLPHF. It’s a standard way above their capacity.

Page 17 Line 296-302: Availability of antimalarial medicines and diagnostics are discussed as reassuring. The discussion around capacity to provide care for uncomplicated malaria needs to be expounded beyond availability of supplies and knowledge. Knowledge and availability of medicines are not the only determinants of provision of quality care. Previous studies have shown that actual practice in the private sector falls below standard, mainly attributed to profit driven mal-practice. Most people are unable to afford the quality of care these facilities provide, with providers compromising (under dosing, providing treatment without testing) on care in the interest of profiteering.

Page 17 Line 307: The word however is misplaced. The sentence is not contrary to the preceding one.

Page 17 Line 308: The problem of irrational antibiotic use is applicable to antimalarial use as well. Additionally, the authors have neglected the concern of spread of AMR with respect to antibiotics and antimalarials. The discussion is incomplete without this point.

Page 17 Line 309-310: The argument suggests that LLPHF should be resuscitating patients in critical condition prior to referral. However, this expectation should be weighed against their registration status and expected standard in Uganda. Is there a policy statement in Uganda (or by the WHO) for management of childhood illness that recommends resuscitation of patients with Oxygen at LLPHF before referral? This discussion point is punching above the weight of LLPHF in Uganda.

The entire discussion needs to be contextualized to Ugandan standards based on existing guidelines and or protocols, for e.g. IMCI. Short of that the discussion remains theoretical and its relevance to local context questionable.

Page 18 Line 318: Which supplies required for teaching how to mix ORS were missing? Is it not possible to give instruction on how to mix ORS without the supplies? It’s possible the concern is inability to demonstrate.

Page 18 Line 325: the words “some facilities” undermine the extent of the problem. It was more than 50% which can’t be rated as ‘some.’

Page 18 Line 326-327: First, the statement “…..it’s contraindicated in management of most diarrheal disease in children below 2 years” is inaccurate. Reference # 40: quotes 3 years as the cut off; loperamide for children below 3 years is problematic above 3 years with minimal dehydration may have benefits. Please quote the reference accurately. The discussion leaves out the fact that loperamide could have benefit as indicated in Reference #40.

Page 19 Line 334-342: Like in most sections in the discussion, the authors are extrapolating their findings to discuss matters they did not study. The discussion should focus on availability or absence of what they studied. The implications can be highlighted, but to suggest repeatedly assumed health care worker practice in the absence of supplies is being speculative. In other words, the discussion is full of too many generalizations, diluting the discussion to author opinions other than presentation of facts.

Page 19 Line 342: Specify appropriate equipment and in the context of expectation

Page 19 Line 344: You can’t start a new paragraph with “Furthermore…”

Page 19 Line 346: The sentence starting.. “This is comparable to other studies [45] and may explain the lack…….” is problematic. First, you mention studies but reference 1 study; they should be more. Second, the explain part is another assumption, not backed by study findings. In fact, if the authors were interested they could, using their data and multivariable analytical methods determine factors associated with low knowledge among health workers in LLPHF.

Page 19 lines 351-355: Limitations: Focusing the study in a single district is not a limitation. Yes, the study findings are not generalizable to every setting which was beyond the scope of the study. Importantly, the study sample was representative of Mbarara the selected study setting (the intended design).

Page 20 line 356: “guess work” refers to the product; incorrect grammar leading to unclear communication

Page 20 line 359: “…our survey was as provided…” incorrect grammar leading to unclear communication. Additionally, can the common childhood illness be specified

Page 20 lines 363-369: The conclusion is poorly written, key findings presented vaguely; seems to be a summary of all the findings. “Inadequately trained” (line 365) is a vague statement. “Lacked good knowledge of the illnesses…. (Line 366) is also vague.

Page 20 lines 366-369: The stated knowledge raising interventions including refresher training and provision of job aids is an assumption. Using their data, the authors can analyse for factors associated with ‘knowledge’ and include recommendations premised on findings and not assumptions.

6. PLOS authors have the option to publish the peer review history of their article (what does this mean?). If published, this will include your full peer review and any attached files.

Reviewer #1: No

Reviewer #2: No

Reviewer #3: No

---

## [Author Response · Author response to Decision Letter 0]

17 Aug 2021

Editorial comments:

1. After review of the manuscript, I have no concerns about the quality or clarity of the writing. The majority of the reviewers felt the manuscript was well written. When assessing the stylistic suggestions of Reviewer #3, the authors should use their discretion on whether to modify the language. I consider these edits to be non-essential. In addition, I leave it up to the authors whether they would like to consider major structural changes to the organization and presentation of the results within the manuscript. However, there are some concerns noted about redundancy with constructive suggestions outlined by Reviewer #1. 

One area that must be addressed is the question raised related to the sampling of the hospitals. I recommend the authors follow the suggestions outlined by Reviewer #1 with additional consideration to the comments by Reviewer #3, if applicable.

Response: We thank the editor for these encouraging comments. Please find the point by point response to each of the reviewers’ comments below.

Response: We have uploaded the data as Supporting Information.

Response: We thank the editor for this suggestion. We have moved the ethical statement to the methods section on page10, lines 183 – 187.

4. Please ensure that you refer to Figure 1 in your text as, if accepted, production will need this reference to link the reader to the figure.

Response: Reference has been made to Figure 1 in the text. This is found in the results section on page 16, line 278.

5. Please upload a copy of Figure 2, to which you refer in your text on page 16. If the figure is no longer to be included as part of the submission please remove all reference to it within the text.

Response: We thank the editorial team for this observation and apologize for this mistake. Reference to Figure 2 was in err. We were referring to Figure 1 not 2. This has been corrected in the text as in our response to comment 4 above.

Reviewer # 1

The authors are to be congratulated on this practical study. They conducted a census of quality of care for pediatric illnesses in a large number (n=110) of health facilities in Western Uganda. This information is helpful in highlighting gaps in clinical care. I would be in support of publication of these data, recognizing that they are quite specific to the locale (not generalizable). Nonetheless, this setting is not unique within Uganda and the broader sub-Saharan African healthcare landscape. It is important to highlight ongoing gaps in clinical care; there is a paucity of such data in the published literature.

Response: We thank the reviewer for the kind remarks

Major comments:

1. “sample” versus total population of health facilities in Mbarara District.P9 line 177: “We randomly selected the LLPHFs from the list of all identified LLPHFs by the licensing bodies 178 and local leadership” and P10 line 200: “data were not collected from 30 facilities”.My understanding of the selection of included facilities is as follows: There were a total of 140 facilities (124 registered and 16 identified through local leaders), of which 30 did not provide data because they met exclusion criteria such as hospital, no pediatric services, lab only, etc, leaving 110 facilities. This appears to be the entire set of eligible facilities in the Mbarara District (please clarify if there were more than 140). If that is the case, this is not a sample, but the entire group of facilities (“population”). If I have understood this correctly, then the sample size calculation is not germane, since all sites were included, it is not a sample. Furthermore, the methods state that a random sample was selected, however, this is not a random sample but the entire set of eligible facilities.If I have understood correctly, please delete the sample size calculation and the statement about a random sample, and simply state that all registered and unregistered facilities in the District were included, with exclusions for hospital, no pediatric services, lab only, etc. If this is not correct, then the site selection needs to be explained more clearly.

Response: We thank the reviewer for this observation and suggestion. Much as we calculated a sample size a priori, we collected data from all eligible health facilities after excluding those which did not meet the inclusion criteria. This was because the remaining facilities were almost equal to the sample size we had calculated. As suggested by the reviewer, we have deleted the sample size calculation as it is no longer applicable to our study. In addition, we have deleted the sentence with random sampling from the text and have revised the methods section to reflect this, on page 11 lines 204 to 210. It now reads as: 

 “We line listed 140 private health facilities in the study area using registers from professional regulatory bodies, the Mbarara District health Office and other identified by local leaders. Out of these, six were hospitals and 13 laboratories and were dropped from the list. We visited the remaining 121 facilities out of which four were permanently closed, 6 did not treat children, and, in one consent was not given by the owner. We collected data from all the remaining 110 health facilities.”

2. Knowledge scale not validated

P9 lines 168: “A score below 50% was considered low, 50-70% as average and above 70% as optimal.” This is quite arbitrary. It is not clear what the implications (for quality of nursing care or clinical care) would be for different levels of knowledge. The fact that the scale is not validated and has not been associated with markers of quality of care or patient outcomes needs to be mentioned in the limitations section. What we have here is a scale (number) without an interpretation. P9 lines 169: “Cronbach’s coefficient alpha score of 0.63.” Reliability is moderate. This knowledge score requires further validation and probably refinement of the scenarios, modification or even deletion of some questions, depending on their performance in contributing to a unified construct. Please list this as a limitation in the discussion (scale not previously validated). Different scores among health workers in different cadres was supportive of the validity of the scale, although this did not reach statistical significance.

Response: We thank the reviewer for these observations and suggestions. It is true that we decided on the scores arbitrarily because we did not find any in literature that had been validated, with implications for the quality of healthcare in similar study settings. In addition we used a tool that had not been previously validated. We have included these facts among the limitations as advised by the reviewer on page 20, lines 362 – 366 and they read:

“In addition the tool had not been validated and had a moderate reliability Cronbach’s coefficient alpha score of 0.63. Nevertheless the fact that different cadres of health workers scored different marks, albeit no statistical difference, supports its internal validity to a certain extent. As the next steps, we need to refine this scale so that it can be used in similar study settings.”

3. Most of the text in Results section is redundant (duplicates Tables).

P11-13: The text of the Results duplicates information found in the Tables. This is not necessary and is redundant. Recommend major reduction of the text and just simply referring to the table, without repetition of numbers that can be found in the Table.

Response: We thank the reviewer for this insightful suggestion. We have modified the text and removed the redundant text as much as possible in the results section on pages 11-16.

Minor comments:

4. P7 line 119 “in-charges” is a colloquial term. Prefer “charge nurse” or other formal title.

Response: We have deleted the word in-charge however, since the responsible staff could be any of the cadres of health professionals we have preferred to call them ‘supervising health workers’ 

5. P8 line 148 “gentamycin” spelling error. Correct spelling: “gentamicin”

Response: The spelling of the medicine has been changed to “gentamicin”

6. P9 lines 165-7: “Likert scales were used to grade the responses to the questions on a 1-3 scale of strongly agree, neither agree nor disagree, and, strongly disagree (Appendix 1).”

This is unusual for a knowledge items which can be scored as correct or incorrect. Likert scales for gradation of agreement seem less applicable here (better suited for "attitudes"). How a response of “neither agree nor disagree” was scored? (I presume incorrect).

Response: We used Likert scales so as not to make the interviewee feel as if they were doing a written examination and not to feel compelled to choose between agree and disagree when they were not sure of the answer. Such situational judgement tests (SJT) of knowledge have been used as proxy indicators for performance (Matošková Jana. Measuring Knowledge. Journal of Competitiveness Vol. 8, Issue 4, pp. 5 - 29, December 2016 ISSN 1804-171X (Print), ISSN 1804-1728, DOI: 10.7441/joc.2016.04.01. 

Indeed a response of “neither agree nor disagree” scored zero.

7. P11 line 213: “rural areas vis-à-vis urban areas” should be “versus”

Response: We thank you for this suggestion. The word has been changed to “versus”. This is now found on page 12, line 221.

8. Tables 3, 4: A study with n=110 cannot measure proportions with such precision (3 significant figures are reported, implying a precision beyond that which is possible with this sample size, +/- 5%, approximately). Please round all percentages to 2 significant figures (as in Table 5 – this one is great!). The P-values are also excessively precisely reported with 3 significant figures – recommend rounding to 2 sig figs.

Response: We agree with the reviewer on this point. We have rounded off all percentages and P-values in tables 3, 4 and 5 to 2 significant figures.

9. Figure. Regarding the number on top of each error bar: the number of participants (or n/N) would be more helpful than providing the percentage (which can be read off the y-axis). Providing the number of participants also helps to understand the precision with which this proportion is estimated. Could consider illustrating this on the graph with error bars, representing the binomial 95% confidence interval on the proportion 70 for example.

Response: We thank the reviewer for this suggestion. We have revised the figure to include the error bars and also indicated the denominators for each cadre of the health worker, as suggested by the reviewer.

10. The literature review for the Discussion is acceptable, but I am aware of some studies (e.g., on oxygen availability in Uganda and elsewhere) that would serve as good comparator studies, and might be considered for inclusion in the discussion:

• Otiangala D, Agai NO, Olayo B, Adudans S, Ng CH, Calderon R, et al. Oxygen insecurity and mortality in resource-constrained healthcare facilities in rural Kenya. Pediatr Pulmonol. 2020;55:1043–9.

• Nabwire J, Namasopo S, Hawkes M. Oxygen availability and nursing capacity for oxygen therapy in Ugandan paediatric wards. J Trop Pediatr. 2018;64(2):97–103.

• Belle J, Cohen H, Shindo N, Lim M, Velazquez-Berumen A, Ndihokubwayo J, et al. Influenza preparedness in low-resource settings: a look at oxygen delivery in 12 African countries. J Infect Dev Ctries. 2010b;4(7):419–24.

Response: We thank the reviewer for these references and suggestion. We have added the references to our discussion section on page 18, lines 322 – 326. It reads:

“This finding is consistent with that of other studies in Uganda and other sub-Sahara African countries which describe inadequate oxygen availability in both public and privately owned health facilities often because of lack of, or, non-functional equipment or unreliable power [37-39]. This contributes to poor outcome in treatment of pneumonia.”

Reviewer #2: 

General comment

Well-written manuscript on a topical issue affecting health care delivery in Uganda. The availability of essential medicines in the low-level health facilities is reassuring .The purely descriptive approach however adds little to the knowledge as to why there was low knowledge among the health workers. Authors need to clarify several statements especially in the results section.

Response: We thank the reviewer for the kind remarks. We have clarified the statements in the result section as suggested by the reviewer. 

1. Abstract

The abstract is well written and conclusions are drawn from the data presented.

Response: Thank you

2. Introduction and background

Line 69: The package was intended to be delivered by both public and private health facilities as it was developed to cover all levels of health care even though lower level facilities lagged behind [13-15]. Sentences need to be re written for clarity. Package meant for both public and private facilities, also developed for all levels Lower levels have lagged behind’ – In doing what? Since when? At this point, a definition of levels of health care facilities in Uganda would help the readership understand lower level facilities are. 

Response: We thank the reviewer for this observation. We have modified the sentences on page 4 lines 66 to 73 to and now read as follows;

“In the year 2000, Uganda implemented the Uganda National Minimum Health Care Package (UNMHCP), in order to provide healthcare for the whole of its population. This was a basic package of essential healthcare services consisting of interventions against the most prevalent diseases in the country such as malaria, HIV/AIDS, diarrheal diseases, perinatal and maternal conditions to mention a few [12, 13]. The package was intended to be delivered by both public and private health facilities at all levels of healthcare provision. Lower-level private health facilities are often left behind during dissemination and training on new policies and guidelines and for this reason have lagged behind in implementing such policies [13-15].”

In addition, we had defined the levels of health care in Uganda in the study settings section on page 5 line 99 and would prefer not to move it.

Line 79: For example, one study reported largely inappropriate management of common childhood illness at private sector drug shops in rural Uganda, while another study found no difference in the management for sick newborns in rural public and private facilities in eastern Uganda [22, 23]. Another study in central Uganda reported great gaps between urban and rural private facilities with rural facilities having less trained health personnel and less availability of some medicines” How comparable are drug shops and say lower level clinic?

Response: We thank the reviewer for this question. Drug shops are usually small walk-in health stores selling over-the counter medicines with a prescription from a health worker or as desired by the patient (Mayora C, Kitutu FE, Kandala N-B, Ekirapa-Kiracho E, Peterson SS, Wamani H. Private retail drug shops: what they are, how they operate, and implications for health care delivery in rural Uganda. BMC health services research. 2018;18(1):532.). Ideally, these should register with the National Drug Authority and should be operated by a person with training in pharmacy or another clinical discipline such as nursing. In most cases they register as drug shops but in actual sense provide more services than only selling and dispensing medicines. In such cases therefore they may be referred to as ‘clinics’ by the users. In our study however, we did not include facilities which only sold and dispensed medicines (page 7, lines 117 – 118). 

3. Justification for doing the study – varying information about the capacity to care for children in private and public health care facilities- The authors need to clarify the justification for a study in lower level facilities in Mbarara district. Could a multi center study – multi region study yielded more meaningful and more generalizable information? How different is Mbarara district from the districts in central Uganda where earlier studies were done?

Response: We thank the reviewer for this comment. We agree with the reviewer that a multi-centre, multi-regional study would have been better in yielding generalizable results. We could not carry out such a study because of financial limitations. We however believe that the results of our study are an important contribution to literature especially since most studies had been done in the central region which has more urban centres and therefore more health workers with higher professional qualifications. Worldwide, urban health facilities have more qualified health workforce than rural facilities. (1. Roger Strasser, Sophia M. Kam, and Sophie M. Regalado. Rural Health Care Access and Policy in Developing Countries. Annu. Rev. Public Health 2016. 37:395–412. doi :110.1146/annurev-publhealth-032315-021507. 2. WHO. Increasing access to health workers in remote and rural areas through improved retention http://apps.who.int/iris/bitstream/handle/10665/44369/9789241564014_eng.pdf;jsessionid=3EE62DD0E5033C176560100DCE1CE489?sequence=1).

In addition, being near the capital city, facilities in the central region benefit more from training and capacity building workshops than far placed like in Mbarara district. We therefore hypothesised that lower-level facilities in Mbarara had lower capacity than those described from facilities in central region. These results hypothesise could mirror other rural districts that are far placed from the capital city. We have added the following statement to out justification on page 5 lines 90 – 95:

“Mbarara district is representative of the country; majority of the population is rural and the district has a homogenous distribution of health facilities with urban clinics and really rural clinics. A confidential inquiry into maternal and child deaths carried in out in two counties of Kashari and Rwampara in 2015 revealed child deaths exceeding the national figures. Majority of the health facilities are privately owned.”

3. Methods

Assessment tools for knowledge – Could it be possible that some of the respondents got the answers correct purely by chance or guess work? How did the authors control for this? Would perhaps self-rating scales or focus group discussions or reflective practice sessions have added value to the assessment tool?

Response: We thank the reviewer for this observation. Indeed we noted this as limitation for this paper. However we used Likert scales with 3 options, strongly agree, neither agree nor disagree and strongly disagree. We hope that providing 3 options would limit guess work by adding the ‘neither agree nor disagree’ option. This would allow the interviewee to reason.

We intend to improve this knowledge assessment tool as follow-up to this work and we shall adapt these insightful reviewer suggestions.

4. Results

i). Data from 110 out of the 140 private listed facilities, 124 of which were registered by regulatory bodies and 16 identified by local leaders, were analysed. This statement is confusing, how many health facilities were studied? How many had complete data? How many facilities had their data analyzed? ‘Out of these, data were not collected from 30 facilities; six of them were hospitals, four facilities were permanently closed, 13 facilities only offered laboratory services, and seven that did not treat children. Out of which ones? The 110, 140 or 124?

Response: We have clarified this information in the health facility selection section on page 10 lines 183 -188. We have deleted most of this information from the results section it now reads;

‘Data from 110 LLPHF were analyzed. We also analyzed data from 129 healthcare workers for knowledge assessment.’

ii). The majority of the health facilities (95%; n=106) of 110 LLPHFs were of a level below HCII of the ministry of health facility classification; three were HCII and three at HCIII level. The numbers do not add up, if n=110, and 106 were below HCII, and then the balance should be 4, and not 6 health centers as shown by the authors numbers.

Response: We thank the reviewer for this observation. It was a transcription error. We have corrected the figures as seen on page 11, lines 221 – 222:

‘The majority of the health facilities (95%; n=104) of 110 LLPHFs were of a level below HCII of the ministry of health facility classification, three were HCII and three at HCIII level.’

iii). Overall, but more stocked by urban (54%) compared to rural facilities 222 (25%), P=0.002. Nearly all LLPHF had paracetamol (97%) and ibuprofen (95%) for fever management- Would be nice to add the numbers (n/N) to add meaning to the percentages.

Response: We thank the reviewer for this suggestion. We have in fact removed all the numbers following advice from another reviewer to avoid redundancy since all these figures are provided in the Tables 2, 3, 4 and 5.

iv). Availability of medical equipment

Impressive results – Were the authors looking out for availability or the use or functionality of these equipment at the health facilities?

Response: We specifically looked out for availability of functional equipment. We have added the word ‘functional’ in the subheading on page 15 line 279.

Reviewer #3: 

General comment

This is an important paper; reflecting on the preparedness and readiness of private health facilities in Mbarara to provide care to children with common childhood illnesses. The role of the private sector in providing health services is a topical area, with their exact role in complementing the public health service delivery system ill defined. The findings of the paper are undoubtedly a valuable contribution to the debate. However, despite its importance the paper is poorly written, methods lacking, results shallow and poorly presented, and the discussion lacks in depth and is premised on too many assumptions rather than actual study findings. Provided below are comments.

Response: We thank the reviewer for acknowledging that this is an important paper and the value of the findings. We note the reviewers concern about the writing style, and the depth of methods and result presentation. The objective of this paper did not require complex analyses and we have presented the results as clearly as possible. Nevertheless, following the reviewer’s comments we have made changes were necessary as specified below, along the reviewer’s specific comments.

Specific comments

Title: The title could be more specific; specify the location where the study was done; Mbarara district. 'Western Uganda' is too wide.

Response: We thank the reviewer for this suggestion but we prefer the leave title as it is. We feel Mbarara District is specific enough since we indeed collected data from the whole district. We know that not everyone who will read the paper knows where Mbarara District is, that is why we have added ‘Western Uganda’.

Key words (Page 3 line 48): should appear on the title page, not after the abstract

Response: We have deleted the key words that were below the abstract and will add these during the online submission so that they appear on the title page.

Abstract (Page 2)

The presented results should match those in the main body of the manuscript and in the tables. They seem to be discrepancies

Response: We thank the reviewer for this comment. We have gone through the results and confirm that the results which appear have in the abstract matches what is in the tables. We summarized these so as not to exceed the 300 word limit set by the journal.

INTRODUCTION

General comments

― For purpose of clarity, the authors should re-write the entire introduction paying attention to spelling, punctuation, and grammar.

― The justification of the study is not well captured in the introduction

Response: We thank the reviewer for this suggestion. We have revised the spellings, grammar and punctuations where necessary and these appear as track changes. In addition we have added text to improve the justification of the study. Apart from the sentence on page 4, line 80 that reads ‘Varying data exist on quality of care for children in private facilities including their capacity to manage childhood illnesses’ we have added the following:

‘Mbarara district is representative of the country; majority of the population is rural and the district has a homogenous distribution of health facilities with urban clinics and really rural clinics. A confidential inquiry into maternal and child deaths carried in 2 counties of Kashari and Rwampara in 2015 revealed child deaths exceeding the national figures. Majority of health facilities are privately owned’

Specific comments

Page 3, line 52: The first paragraph of the introduction starts with a complex sentence. The sentence is: 1) too long, 2) comprised of too many clauses, and 3) starts with a subordinate clause, the second clause being the main idea. For purposes of scientific writing and clarity it may be best to start the sentence with the main clause. Use of a subordinating conjunction ‘As’ maybe grammatically correct for a sentence, but is not the best way to start a paragraph.

Page 3, line 55: The 2nd sentence, like the preceding one starts with ‘As’ Using ‘As’ repeatedly (appears multiple times in the introduction) makes the reading repetitive.

― Revise the 1st and 2nd sentences of the first paragraph. Good to start with clarity and to the point.

Response: We have revised the sentences following the reviewer’s comments and they now read as follows (page 3, line 52 – 57):

‘The Universal Health Coverage (UHC) is one of the targets of the Sustainable Development Goal 3(SGD3). It aims at improving health outcomes through expanding access to essential healthcare services for all people in need while minimizing risk of encountering excessive financial hardships. Many African countries are still lagging behind, in achieving the UHC target yet the year 2030 is approaching.’

Page 3, line 56: The 4nd sentence starting “In 2019, Sub Saharan Africa was reported to be responsible….” is grammatically incorrect. SSA cannot be responsible for death, but may have accounted for….

Response: We have modified the sentence and it now reads:

‘In 2019, Sub-Saharan Africa accounted for half of the over 5 million deaths which occurred globally in children below 5 years of age.’

Page 4 line 65: The sentence starts with a conjunction ‘in order’ again. The sentence should start with the main clause, thus: In 2000, Uganda implemented……… Additionally, in the same sentence (line 67) the words ‘disease conditions’ sound repetitive, either word would have sufficed

Response: We have modified the sentences and they now read as follows (page 4, lines 66 – 70):

‘In the year 2000, Uganda implemented the Uganda National Minimum Health Care Package (UNMHCP), in order to provide healthcare for the whole of its population. This was a basic package of essential healthcare services consisting of interventions against the most prevalent diseases in the country.’ 

Page 4 line 70: The sentence starting “In the rural areas of Uganda…” implies private facilities are more accessible within the recommended 5km. How about outside the 5km?

Response: We have noted the reviewer’s concern however we have decided to keep this sentence on page 4 lines 73 -75 as it is. The aim of this point was to emphasize that private health facilities are more accessible compared to public health facilities. We did not think it necessary to discuss about facilities outside the 5km radius.

The authors use the words services and care interchangeably, there is need to be consistent with choice of words expressing the same point.

Response: We thank the reviewer for this observation. We replaced the word ‘service’ with ‘health care’ in the text on page 4, line 78.

METHODS

General comments

― For clarity, include more specific sub-sections avoid merging different subjects in the same sub-title

― Include the following subtitles: study design, identification and selection of facilities,

Page 5 line 91: Separate the sub titles, study setting and study population. Comment on the level of development in Mbarara and the social economic status of the population

Response: We have added a sub title on page 5 line 97, for study design but combined it with study population. This is because study design is a very short statement that we think should not warrant a whole section dedicated to it. We have combined the two phrases on design and population to make one sentence which reads as follows:

‘We conducted a cross-sectional study among low-level private health facilities in Mbarara District in southwestern Uganda between May and December 2019’

We also added a statement on the socioeconomic status of the district as follows:

‘Mbarara district has a mainly rural population estimated at 473,000 inhabitants, residing in 742 villages, 83 parishes, 16 sub-counties and 3 counties. Fifty three percent of the households depend on subsistence farming for their livelihood.’

Table 1 is not necessary, a summary of the structure can be provided in text.

Response: We thank the reviewer for the comment but we prefer to keep this table as it summarizes the health facility classification for Uganda. We believed the text would be too heavy. We therefore like to keep Table 1. 

Page 5 line 94: The sentence starting “The health facilities in the district follow the Ugandan…….” Did the facilities really follow the Uganda health system? Use of the word follow as a verb is inappropriate.

Response: We have modified the statement to read as follows:

‘The level of health facilities in the district is according to the Ugandan health system structure…’

With respect to health care system at the district level, mention: 1) decentralized health system, and 2) the HSD a functional division of the district health system, equivalent to a constituency/county. Service delivery in the HSD is provided at outpatient facilities tiered at three levels 2 (parish), level 3 (sub county), level 4 (county). This system is applicable to public health care system, what about private sector? Is it the same or different? Please elaborate the structure (types and levels) of the private health care providers.

Response: The levels for the private health facilities too are set according to the ministry of health guidelines and this is considered when the different professional regulatory bodies are determining the licensure fees. We have added a statement to this effect on page 6 line 108 – 109.

Page 6, lines 102-103…the statement, “….the remaining 118 are private clinics and nursing homes which are below HCII level in reference to the MOH guidelines.” This means these facilities were level 1-health centers. The facilities are registered by councils; during registration the facility type and level should have been defined. Were these details available, Please clarify?

Response: We have clarified the statement as follows:

‘The remaining 118 are private clinics and nursing homes which with structures less than HCII level in reference to the ministry of health facility classification, but higher than small drug shops.’

Page 6, line 110: the definition of LLPHFs is vague. First, use of the word 'equivalence' is redundant, as some of the facilities were already defined as either level II or III facilities. Second, for those that didn’t have a level how was equivalence to level II/III determined? What criterion was used?

Response: We rephrased the definition on page 7 lines 124 – 126 as follows;

‘For this study, we defined low-level private health facilities (LLPHFs) according to the Uganda Ministry of Health (MoH) health facility classification as those private facilities at HCIII and below, offering ‘first-point-of-care” services to children’

Page 6, line 116: How were drug shops distinguished from other private providers. It should be noted that the potential for overlap between drug shops and private clinics is high. It’s possible that drug shops were captured as clinics. Please elaborate on how this was minimized.

Response: We mentioned in the preceding sentence on page 7 lines 127 to 130 that they were included as long as they provided health care more than selling or dispensing medicine.

Page 7, line 125: Data collection instruments

― The write up on data collection instruments is disorganized and sloppy. For clarity, list the questionnaires and for each state: 1 the purpose of the questionnaire, 2) who administered the questionnaire, and 3) who the respondent was.

― It’s stated (Page 7 LINE 125) “….self administered questionnaire to assess capacity of LLPHF. This is misleading; later on (page 7 line130 -132) it appears that self administered questionnaire were used to specifically assess KAP

― Line 130 The statement “10 vignettes with 50 questions.” is ambigous. Does it mean each vignettes was comprised of 50 questions, or all added up to 50 questions. Would be best to state how many questions per vignettes. While describing the vignettes you in-appropriately referred to them as knowledge questionnaires, which is a bit confusing. Were the questions focused on specific diseases, if yes, please indicate them

Response: We thank the reviewer for these observations and following the reviewers insightful suggestions we have modified various statements on page 8 lines 139 -149. These are shown as track changes.

Page 8-9: Study variables

― The WHO categorizes outcome measures for health facility assessments in terms of ‘service availability’ and ‘service readiness.’ The authors need to ascertain if this was a 'service availability' and ‘service readiness’ survey. Use of the term ‘capacity’ undermines the scope of work done. The term capacity of the health facility should not be mixed up with health workers qualification and knowledge

Response: We thank the reviewer for this suggestion. Our work falls under the WHO service readiness category. Indeed service reediness is defined as the capacity of the health facilities to provide health services (https://cdn.who.int/media/docs/default-source/service-availability-and-readinessassessment(sara)/sara_overviewpresentation.pdf?sfvrsn=ce58374f_3). We would prefer to remain modest with our work because due to logistical difficulties, we did not collect information for all domains to assess full service readiness. We, however collected more items in some domains than is stated in the WHO’s Service Availability and Readiness Assessment (SARA) guidelines. For example we followed the Uganda Clinical guidelines to collect information on medicines, which are more than those listed as tracer medicines for Child health service readiness (https://www.who.int/data/data-collection-tools/harmonized-health-facility-assessment/introduction). We added assessment of knowledge of the health workers because we did not want to assume that they had adequate knowledge. This, more so because majority of the health workers were of a low cadre. We have incorporated some of the terminology used by WHO’s Service Availability and Readiness Assessment (SARA) to improve our manuscript presentation but would prefer to keep the variables as they are.

― Page 8, line 144-145: How does lack of vital medicines cause side effects?

Response: This is the definition given by the Uganda Clinical Guidelines. We think in the context it used, ‘side effects’ means unfavourable outcomes when the medicine is not given (Uganda clinical guidelines 2016; national guidelines for management of common conditions. Kampala: Ministry of Health Uganda).

― Page 8, line 151-152: Facilities should have been assessed based on expected capacity. Assessing facility performance against standards above their ability is meaningless. For example, do we expect lower level facilities to be providing Oxygen and blood transfusion service? Probably not, and if they did, that would translate to be a problem.

Response: We note the reviewer’s concern. While we did not expect most of these facilities to have oxygen and facilities for blood transfusion we went ahead to look out for facilities that had these services because we expected private facilities at HCIII level to have them. Oxygen is an important first line of resuscitation for children with severe disease and therefore a facility having it is of added value. HCIII should actually have blood transfusion services because they admit and deliver pregnant mothers. Not having oxygen nor blood transfusion services would not translate into a problem but having these lifesaving services would imply added value of that private facility. 

― Page 9 Line 164-165: the description of the vignettes is more clear than was on Page 7, line 140

Response: This has been noted. 

― Page 9 Line 167-165: All questions carried equal weight…should be written as 'the correct answers to all questions carried equal weight'

Response: We thank the reviewer for the suggestion, the sentence on page 10, line 186 has been modified accordingly.

― Page 9 Line 168-169: What was the basis of the performance cut offs.

Response: We did not have a reference for cut offs so we used arbitrary figures. 

Page 9 line 165-168: Can the authors provide more detail on how they analyzed (scored and summarized) data collected using the likert scales. The scales comprised of 3 options, agree, neither agree or disagree and disagree. How were right and wrong answers handled? and what was the overall distribution of responses based on the 3 options.

Response: More details have been added page 10, lines 186 -190 to explain the knowledge scores. It read as follows:

‘Strongly agree and strongly disagree scored one point depending on whether it was the correct alternative. Neither agree nor disagree scored zero. All the correct answers to all questions carried equal weight so a simple summation was employed to get the total score out of 50 marks and percentage calculated. A score below 50% was considered low, 50-70% as average and above 70% as optimal.’

Page 9: Sample size and sampling

― Create an independent sub-section for sampling. What sampling procedure was used to select facilities? Why did you sample one or two health workers; was a representative sample obtained

Response: A sub-section for sampling, now called facility selection, has been created on page10 -11, line 199 -205. We made a line list of all possible private health facilities as detailed in the manuscript and surveyed all facilities that fitted the selection criteria. For health workers, we purposively selected those who treated children. Since the facilities usually have 1 -2 such staff at any one time, we aimed to interview at least one and a maximum of 2 from each facility to cover all the facilities. 

Results

Page 10 Lines 200-203. The first and second sentences are poorly written and confusing. The order of sequencing the events leading to enrollment of facilities that were studied is not logical. A study profile highlighting how facilities were selected from the entire pool would be helpful.

Response: This section has been re-written. Most of the information concerning the number of facilities studied has been transferred to the facility selection section, page10 -11, line 199 -205.

Page 10 Lines 204: The sentence starting “The majority of the health facilities (95%; n=10) of 110…. ” is poorly written. Start with the overall state. Thus: Of the studied LLPHFs, most (106; 95%) were below level II………….

Response: This sentence has been modified following the reviewers suggestion. This is on page 12, lines 237 – 240.

Page 10 Line 204-206: Of 110 facilities, 106 were below level II, three were level II, and three were level III giving a total 112 facilities greater than the number of facilities (110) enrolled. Where did the extra two come from? Correct this error

Response: We apologize for this discrepancy. It was a transcription error and has been corrected. Thank you.

Page 10 Line 206-207: Urban and rural…how were these defined and determined? Mbarara city is urban; its surroundings are peri-urban. Therefore, facilities in peri-urban areas could have been captured as rural. Kindly clarify

Response: The definition of urban vs rural facilities is on page 11 lines 223- 225. We considered urban facilities as those found within the current day Mbarara City. We divided the facilities into 2 categories only. We hope this point is clearer now.

Include a sub-title ‘Baseline characteristic of health facilities and health care workers’ Baseline results of facility and health care workers should be reported under this section. Other characteristics to consider in this section include:

― Health care worker age, gender, and year of training.

― Facility ownership and or registration status

Replace table 2 with a table of baseline characteristics combining both facility and health care worker characteristics.

Response: We thank the reviewer for this suggestion. The sub-title ‘Baseline characteristics..’ has been created. Our study was descriptive in nature and the results are meant to describe the capacity of the LLPHFs in terms of availability of human resource, supplies and infrastructure. We did not collect data on some of the variables suggested by the reviewer and are thus unable to present these additional data. We therefore prefer not to change Table 2.

Page 11 Availability of medicines; The word ‘overall’ is used repeatedly; it is best placed at the start of the sentence.

Response: This section has been modified following this reviewer and another reviewer’s suggestion. Most of the information has now been removed from the text and is presented in table 3.

Page 11 Line 217: replace pneumonia with ‘non-severe pneumonia’

Response: This sentence was deleted in a bid to reduce on the redundancy as advised by another reviewer.

Page 11 Line 218: include the results for availability of antibiotics in rural and urban facilities.

Response: As above, this is information is presented in the table. We have not added it to the text to avoid redundancy.

Page 11 Line 219: Parenteral ampicillin is referred to in this sentence; however, it’s not captured in Table 3. Additionally, why are the results for parenteral antibiotics (ampicillin, penicillin, gentamycin) combined? They should be presented independently.

Response: These were deleted from text as above to reduce redundancy.

Page 12 Line 231: Table 3. In addition to urbanization level, results should be stratified by facility level and registration status. Important differences could arise within these sub-groups

Response: We thank the reviewer for this suggestion. In the data management and analysis section on page 12, lines 230 - 233, we justified the reason for stratifying our analyses by rural vs urban based on previous studies that have been conducted elsewhere in Uganda. It reads:

‘We stratified our analyses by location of the health facilities (rural vs urban) because of the previously observed discrepancies in the capacity between private health facilities elsewhere in the country, and to better inform possible targeted interventions.’’

Since we do not have a scientific basis to stratify by the different levels of the LLPHFs, we prefer to keep our analyses as they are. We believe that these results are informative, with regard to the study objective.

Page 11 Line 222: The result presented for Artemether/Lumefantrine is different from that in Table 3. Same problem applies for the results of all others medicines presented in the text; they don’t match the results presented in Table 3. Please correct for consistency.

Response: We thank the reviewer for this observation. The figures have been corrected so that any figure mentioned in the text matches what is presented in the tables.

For comparison of % across urban vs. rural facilities, can the absolute difference be included with the 95%CI around the difference.

Response: We thank the reviewer for this suggestion. We presented the numbers and proportions in the tables and compared them across rural and urban health facilities, using Chi square. The corresponding p values are presented in the tables as well. We believe this information is adequate for the readers to compare the capacity of the rural and urban health facilities. We therefore prefer to keep the tables as they are, since adding absolute differences adds no additional benefit, and may potentially congest the tables.

Page 13 Line 234 Availability of equipment and other medical supplies. The proportions presented in this section, do not match those in Table 4; they should be the same.

Response: Most figures have been removed from the text as they are presented in the table to avoid redundancy as explained in the above.

Page 14 line 251: Table 4. Can all the p-values be re-calculated? Some seem to be wrong. For example, adult weighting scale; the significance of difference between urban and rural proportion returns a p-value of 0.053 and not 0.032

Response: We thank the reviewer for pointing this out. We have cross-checked all the p values and confirm that they are correct. In table 4, we indeed have the correct p value of 0.032, and not 0.053, as the reviewer pointed out. 

Page 14 line 251: Table 4. Like table 3, results should be stratified by facility level and registration status. Important differences could exist between these sub-groups

Response: We thank the reviewer for this suggestion. However, as noted in our methods, the sampling for the health workers who were assessed for knowledge, and selection of the health facilities were different. As such, we used two entirely different databases and tools. We are unable to link the two databases at this point so as to carry out these additional analyses requested by the reviewer.

Page 14 line 255: Infrastructure and availability of trained human resource

The title could be made more specific ‘Availability of infrastructure and healthcare workers trained in IMCI’

Response: We thank the reviewer for this suggestion. We have changed this sub title accordingly.

The order of presentation of results should match the order in the title. Therefore start with infrastructure. The order in which the results are presented in the sub-section does not match the order of presentation in Table 5

Response: The order of mention has been changed in the titles to match the order of presentation in the text and table. 

Page 14 Line 256: The first sentence states, “Of 110 facilities, 14 (13%) had at least one doctor trained on IMCI…….as shown in Table 5. However, in Table 5, the category for at least one trained doctor does not exist. Additionally, for nursing assistants the category 'Two or more' is missing

Response: We are sorry for the confusing way in which we wrote these results. This information is in Table 5 indeed. We had summed up (10+4=14) facilities that had one or more doctors trained. As shown in the table, the remaining 87 facilities did not have any trained doctor. However, in response to comments from other reviewers, we have removed most of this redundant narrative and replaced it with a more summarised statements. We hope this is no longer confusing. It now reads (page 17, lines 303-304):

‘Only a few of the health facilities had health workers who had ever trained in IMCI as is shown in Table 5. This finding was similar for both rural and urban health facilities.’

Page 15 Line 261: Table 5, What is the importance of the order of ranking for the number of health care workers: “None,” “One,” and “Two or more.” Does being trained two or more times matter more than being trained once? What would have been more important is the duration since last training. Additionally, why only focus on IMCI training, how about other trainings, why were these not considered?

Response: We thank the reviewer for this question. We hypothesised that ranking is important because we believe repeated training reinforces the information. We therefore believe the more number of times a health worker is trained the better the knowledge and skills. In addition, the information passed on may be a little bit different as new information becomes available. The period we considered for all the training was in the last 6 years. We concentrated on training in IMCI because the IMCI guidelines are meant for health facilities at the primary level and encompasses all the disease conditions we targeted in this study.

Page 15 Line 261: Table 5, for the characteristic ‘Examination room with appropriate lighting’ the value for Urban (32) and rural (44) don’t add up to the total (69)

Response: We thank the reviewer for this observation. We have revised and corrected this error. The correct value is 76 (69%) in Table 5.

Page 15 Line 264: First sentence “….about 63%....” is sloppy writing, ‘about’ suggests uncertainty. In previous sections results are presented as an absolute number with % in parenthesis. In this section, results are presented a % only. Need to be consistent.

Response: We have improved the sentence. It is now found on page 18, line312 and has been modified to read as follows:

‘More than half of the facilities had an examination room with appropriate lighting’

Page 16 Line 271: "Upon stratification by health worker qualification......" can the knowledge results be presented positively, i.e. those who passed rather than those who did not fail.

Response: This has been modified following the reviewer’s suggestion.

Page 16 Line 273: Degree nurses are not included in the Figure.

Response: This has been corrected. The degree nurses and doctors where combined because there was only 1 degree nurse. This is now reflected in Figure 1.

Page 16 Line 273: Figure 2 should be Figure 1

Response: This has been corrected in the text.

Discussion

Page 16 Line 284: Lack of oxygen should not be presented as a gap with respect to these LLPHF. It’s a standard way above their capacity.

Response: We agree with the reviewer. We have rephrased the statement to bring out the point in a positive manner that some facilities had oxygen. This puts them a class above the public low-level facilities because they are able to offer the first line of resuscitation to children with respiratory distress. We have added a sentence that reads as follows on page 18, line 3333 -334: 

 ‘Some facilities had oxygen which is not found at facilities of same level in public sector.’

Page 17 Line 296-302: Availability of antimalarial medicines and diagnostics are discussed as reassuring. The discussion around capacity to provide care for uncomplicated malaria needs to be expounded beyond availability of supplies and knowledge. Knowledge and availability of medicines are not the only determinants of provision of quality care. Previous studies have shown that actual practice in the private sector falls below standard, mainly attributed to profit driven mal-practice. Most people are unable to afford the quality of care these facilities provide, with providers compromising (under dosing, providing treatment without testing) on care in the interest of profiteering.

Response: We thank the reviewer for this suggestion. Indeed this observation is true and in a paper we published recently we found inappropriate care for malaria in the private health facilities. With the reviewer’s advice we have modified this point and it now reads (page 20, lines 349 -353);

‘In reality, this would only be possible if the other conditions for provision of quality care are in operation, including health workers with adequate knowledge on management of malaria. A recent article by our research team described inappropriate care for malaria and other common paediatric infections, implying that it requires more than availability of medicines and diagnostics.’

Page 17 Line 307: The word however is misplaced. The sentence is not contrary to the preceding one.

Response: The word ‘however’ has been replaced with ‘in reality’.

Page 17 Line 308: The problem of irrational antibiotic use is applicable to antimalarial use as well. Additionally, the authors have neglected the concern of spread of AMR with respect to antibiotics and antimalarials. The discussion is incomplete without this point.

Response: We thank the reviewer for this comment. We had intimated to this point of AMR in our discussion on irrational antibiotic prescriptions. Following the reviewer’s advice we added a few words page 20 lines 361 – 364 to make the statement complete. The whole statement now reads as follows:

 ‘Available essential antibiotics are only useful when used in a responsibly. However, studies from countries at all income levels have shown that antibiotics are often used irrationally which contributes unnecessarily to antibiotic resistance [37-39]. The same is also true for antimalarials.’

Page 17 Line 309-310: The argument suggests that LLPHF should be resuscitating patients in critical condition prior to referral. However, this expectation should be weighed against their registration status and expected standard in Uganda. Is there a policy statement in Uganda (or by the WHO) for management of childhood illness that recommends resuscitation of patients with Oxygen at LLPHF before referral? This discussion point is punching above the weight of LLPHF in Uganda.

Response: We thank the reviewer for this comment. We included this statement because the range of the LLPHF is wide, including facilities at health III level. In addition, some of these facilities are operated by or employ doctors who should be able to give the first line of resuscitation to critically ill children before referral for escalation of care.

The entire discussion needs to be contextualized to Ugandan standards based on existing guidelines and or protocols, for e.g. IMCI. Short of that the discussion remains theoretical and its relevance to local context questionable.

Response: We thank the reviewer for this comment. We have discussed the paper in the context of Uganda often referring to country guidelines and similar research carried out in the country but also referring to universal guidelines and global context. 

Page 18 Line 318: Which supplies required for teaching how to mix ORS were missing? Is it not possible to give instruction on how to mix ORS without the supplies? It’s possible the concern is inability to demonstrate.

Response: We thank the reviewer for this suggestion. Indeed, we meant demonstration of how to prepare ORS to ensure that the correct concentration, and, the use of clean utensils and clean water. It is also important to show the caretaker how much ORS to give and how to give. Indeed in the IMCI guidelines, it is recommended for facilities to have a ORS corner where the demonstration takes place. The materials required for demonstration include packets of the ORS, a source of clean water, clean cups for measuring and administering, a spoon, a jar. The wording of the sentence have changed and it now reads as follows:

‘However, the supplies for demonstration of how to prepare and administer ORS were available in only 36% of the facilities.’ 

Page 18 Line 325: the words “some facilities” undermine the extent of the problem. It was more than 50% which can’t be rated as ‘some.’

Response: The word some has been replaced with ‘more than half of the’. This is on page 21 line 385.

Page 18 Line 326-327: First, the statement “…..it’s contraindicated in management of most diarrheal disease in children below 2 years” is inaccurate. Reference # 40: quotes 3 years as the cut off; loperamide for children below 3 years is problematic above 3 years with minimal dehydration may have benefits. Please quote the reference accurately. The discussion leaves out the fact that loperamide could have benefit as indicated in Reference #40.

Response: We agree with the reviewer. This is why, we mention diarrhoeal diseases among a specific age group i.e. below 2 years. We combined findindings of references 46 and 47. We have modified our statement by removing the word ‘most’ (page 21, line 387).

Page 19 Line 334-342: Like in most sections in the discussion, the authors are extrapolating their findings to discuss matters they did not study. The discussion should focus on availability or absence of what they studied. The implications can be highlighted, but to suggest repeatedly assumed health care worker practice in the absence of supplies is being speculative. In other words, the discussion is full of too many generalizations, diluting the discussion to author opinions other than presentation of facts.

Response: We thank the reviewer for this comment. We have tried as much as possible to tone down on the language. On page 22 lines 391 - 393, the sentence that read ‘The survey revealed low availability of basic equipment used for assessing children including weighing scales, MUAC tapes, and timers especially in the rural facilities. This may be an indicator that assessment of children is probably not performed completely.’ With a modified sentence that reads; 

‘The survey findings of low availability of basic equipment used for assessing children including weighing scales, MUAC tapes, and timers may imply incomplete assessment of children.’

Page 19 Line 342: Specify appropriate equipment and in the context of expectation

Response: We thank the reviewer for this comment but believe the equipment we mention are indeed basic and expected at this level of health care. These include weighing scales, simple timers, MUAC tapes etc which can be used at primary level facilities. 

Page 19 Line 344: You can’t start a new paragraph with “Furthermore…”

Response: The word ‘furthermore’ has been removed from the sentence on line 399.

Page 19 Line 346: The sentence starting.. “This is comparable to other studies [45] and may explain the lack…….” is problematic. First, you mention studies but reference 1 study; they should be more. Second, the explain part is another assumption, not backed by study findings. In fact, if the authors were interested they could, using their data and multivariable analytical methods determine factors associated with low knowledge among health workers in LLPHF.

Response: This has been changed to ‘This is comparable to another study carried out in Nigeria [51]’. As explained previously, we are unable to perform multivariable analysis with our datasets.

Page 19 lines 351-355: Limitations: Focusing the study in a single district is not a limitation. Yes, the study findings are not generalizable to every setting which was beyond the scope of the study. Importantly, the study sample was representative of Mbarara the selected study setting (the intended design).

Response: We thank the reviewer for this insightful observation. We agree with the reviewer entirely and have deleted this statement from the limitations (page 23, lines 413 – 415).

Page 20 line 356: “guess work” refers to the product; incorrect grammar leading to unclear communication

Response: the word has been changed to “guessing”

Page 20 line 359: “…our survey was as provided…” incorrect grammar leading to unclear communication. Additionally, can the common childhood illness be specified?

Response: The grammar has been corrected. In addition, the illness being referred to have been specified (page 23, lines 421 - 424(.

Page 20 lines 363-369: The conclusion is poorly written, key findings presented vaguely; seems to be a summary of all the findings. “Inadequately trained” (line 365) is a vague statement. “Lacked good knowledge of the illnesses…. (Line 366) is also vague.

Response: We thank the reviewer for this comment but believe that our conclusions are in keeping with the scope of this study and the findings. 

Page 20 lines 366-369: The stated knowledge raising interventions including refresher training and provision of job aids is an assumption. Using their data, the authors can analyse for factors associated with ‘knowledge’ and include recommendations premised on findings and not assumptions.

Response: We thank the reviewer for this comment. We are again limited in how many factors we can assess apart from the cadre of staff. This could be something we can take up for further research.

---

## [Editor Report · Decision Letter 1]

13 Sep 2021

Capacity to provide care for common childhood infections at low-level private health facilities in Western, Uganda

PONE-D-21-18488R1

Dear Dr. Mwanga-Amumpaire,

We’re pleased to inform you that your manuscript has been judged scientifically suitable for publication and will be formally accepted for publication once it meets all outstanding technical requirements.

Kind regards,

Andrea L. Conroy, PhD

Academic Editor

PLOS ONE
---

## [Editor Report · Acceptance letter]

22 Sep 2021

PONE-D-21-18488R1 

Capacity to provide care for common childhood infections at low-level private health facilities in Western, Uganda 

Dear Dr. Mwanga-Amumpaire:

I'm pleased to inform you that your manuscript has been deemed suitable for publication in PLOS ONE. Congratulations! Your manuscript is now with our production department. 

Kind regards, 

on behalf of

Dr. Andrea L. Conroy 

Academic Editor

PLOS ONE